# VLATTACK: Multimodal Adversarial Attacks on Vision-Language Tasks via Pre-trained Models

**Ziyi Yin**[1]  **Muchao Ye**[1]  **Tianrong Zhang**[1]  **Tianyu Du**[2]
**Jinguo Zhu**[3]  **Han Liu**[4]  **Jinghui Chen**[1]  **Ting Wang**[5]  **Fenglong Ma**[1*]
[1]The Pennsylvania State University, [2]Zhejiang University,
[3] Xi'an Jiaotong University, [4]Dalian University of Technology, [5]Stony Brook University
`{ziyiyin, muchao, tbz5156, jcz5917, fenglong}@psu.edu`
`zjradty@zju.edu.cn, lechatelia@stu.xjtu.edu.cn`
`liu.han.dut@gmail.com, twang@cs.stonybrook.edu`

## Abstract

Vision-Language (VL) pre-trained models have shown their superiority on many multimodal tasks. However, the adversarial robustness of such models has not been fully explored. Existing approaches mainly focus on exploring the adversarial robustness under the white-box setting, which is unrealistic. In this paper, we aim to investigate a new yet practical task to craft image and text perturbations using pre-trained VL models to attack black-box fine-tuned models on different downstream tasks. Towards this end, we propose VLATTACK[2] to generate adversarial samples by fusing perturbations of images and texts from both single-modal and multimodal levels. At the single-modal level, we propose a new block-wise similarity attack (BSA) strategy to learn image perturbations for disrupting universal representations. Besides, we adopt an existing text attack strategy to generate text perturbations independent of the image-modal attack. At the multimodal level, we design a novel iterative cross-search attack (ICSA) method to update adversarial image-text pairs periodically, starting with the outputs from the single-modal level. We conduct extensive experiments to attack five widely-used VL pre-trained models for six tasks. Experimental results show that VLATTACK achieves the highest attack success rates on all tasks compared with state-of-the-art baselines, which reveals a blind spot in the deployment of pre-trained VL models.

## 1 Introduction

The recent success of vision-language (VL) pre-trained models on multimodal tasks have attracted broad attention from both academics and industry [1, 2, 3, 4, 5, 6, 7]. These models first learn multimodal interactions by pre-training on the large-scale unlabeled image-text datasets and are later fine-tuned with labeled pairs on different downstream VL tasks [8, 9, 10]. In many cases, these pre-trained models have revealed more powerful cross-task learning capabilities compared to training from scratch [11, 12]. Despite their remarkable performance, the **adversarial robustness** of these VL models is still relatively unexplored.

Existing work [13, 14, 15, 16] conducting adversarial attacks in VL tasks is mainly under the **white-box** setting, where the gradient information of fine-tuned models is accessible to attackers. However, in a more realistic scenario, a malicious attacker may only be able to access the public pre-trained models released through third parties. The attacker would not have any prior knowledge about the

---

*Corresponding author.
[2]Source code can be found in the link `https://github.com/ericyinyzy/VLAttack`.

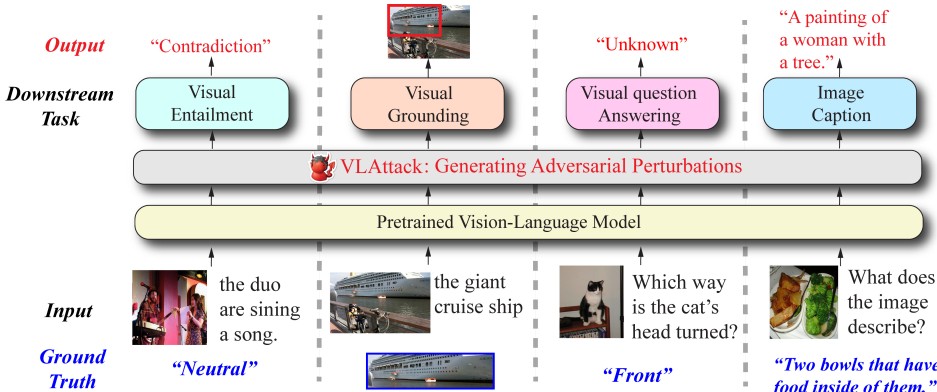

Figure 1: An illustration of the problem of attacking block-box downstream tasks using pre-trained vision-language models.

parameters learned by downstream VL models fine-tuned on private datasets. Towards bridging this striking limitation, we investigate a new yet practical attack paradigm – *generating adversarial perturbations on a pre-trained VL model to attack various **black-box** downstream tasks fine-tuned on the pre-trained one.*

However, such an attack setting is non-trivial and faces the following challenges: (1) **Task-specific challenge**. The pre-trained VL models are usually used for fine-tuning different downstream tasks, which requires the designed attack mechanism to be general and work for attacking multiple tasks. As illustrated in Figure 1, the attacked tasks are not only limited to close-set problems, such as visual question answering, but also generalized to open-ended questions, such as visual grounding. (2) **Model-specific challenge.** Since the parameters of the fine-tuned models are unknown, it requires the attack method to automatically learn the adversarial transferability between pre-trained and fine-tuned models on different modalities. Although the adversarial transferability [17, 18, 19, 20] across image models has been widely discussed, it is still largely unexplored in the pre-trained models, especially for constructing mutual connections between perturbations on different modalities.

To address all the aforementioned challenges, we propose a new yet general Vision-Language Attack strategy (named VLATTACK) to explore the adversarial transferability between pre-trained and fine-tuned VL models, as shown in Figure 2. The whole VLATTACK scheme fuses perturbations of images and texts from two levels:

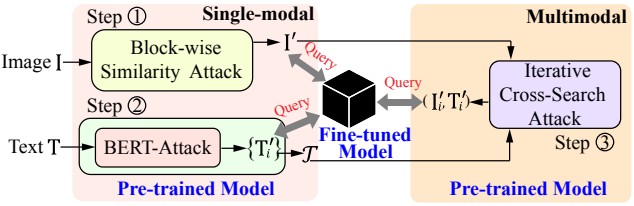

Figure 2: A brief illustration of VLATTACK.

**Single-modal Level**. VLATTACK independently generates perturbations on a single modality, following a "*from image to text*" order as the former can be perturbed on a continuous space. The single-modal attack can effectively detect the adversarial vulnerability of an image or text, and hence avoids redundant perturbations on the other modality. Specifically, to fully utilize the image-text interactions that have been stored in the pre-trained model, we propose a novel **block-wise similarity attack** (BSA) strategy to attack the *image modality*, which adds perturbations to enlarge the network block-wise distance between original and perturbed features in the pretrained model, disrupting the universal image-text representations for the downstream predictions. If BSA fails to change the prediction after querying the fine-tuned black-box model, VLATTACK will attack the text modality by employing the word-level perturbation techniques [21, 22, 23, 24]. We adopt BERT-Attack [21] to attack the *text modality* as its prominent performance has been widely verified in many studies [25, 26, 27]. Finally, if all the text perturbations $\{\mathbf{T}'_i\}$ fail, VLATTACK will generate a list of perturbed samples $\mathcal{T}$ and feed them to the multimodal attack along with the perturbed image $\mathbf{I}'$.

**Multimodal Level**. If the above attack fails to change the predictions, we cross-update image and text perturbations at the multimodal level based on previous outputs. The proposed **iterative cross-search attack** (ICSA) strategy updates the image-text perturbation pair $(\mathbf{I}'_i, \mathbf{T}'_i)$ in an iterative way

by considering the mutual relations between different modal perturbations. ICSA uses a text perturbation $\mathbf{T}'_i$ selected from the list $\mathcal{T}$ as the guidance to iteratively update the perturbed image $\mathbf{I}'_i$ by employing the block-wise similarity attack (BSA) until the new pair $(\mathbf{I}'_i, \mathbf{T}'_i)$ makes the prediction of the downstream task change. In addition, text perturbations are cross-searched according to the semantic similarity with the benign one at the multimodal attack level, which gradually increases the extent of the direction modification to preserve the original semantics to the greatest extent.

Our **contributions** are summarized as follows: (1) To the best of our knowledge, we are the first to explore the adversarial vulnerability across pre-trained and fine-tuned VL models. (2) We propose VLATTACK to search adversarial samples from different levels. For the single-modal level, we propose the BSA strategy to unify the perturbation optimization targets on various downstream tasks. For the multimodal level, we design the ICSA to generate adversarial image-text pairs by cross-searching perturbations on different modalities. (3) To demonstrate the generalization ability of VLATTACK, we evaluate the proposed VLATTACK on **five widely-used VL models**, including BLIP [28], CLIP [29], ViLT [1], OFA [5] and UniTAB [4] for **six tasks**: $(i)$ VQA, $(ii)$ visual entailment, $(iii)$ visual reasoning, $(iv)$ referring expression comprehension, $(v)$ image captioning, and $(vi)$ image classification. Experimental results demonstrate that VLATTACK outperforms both single-modal and multimodal attack approaches, which reveals a significant blind spot in the robustness of large-scale VL models.

## 2  Related Work

**Single-modal Adversarial Attack Methods**. *Image Attack*. Traditional image attack methods [30, 31] generate adversarial samples by optimizing the loss function with regard to the decision boundary from model outputs. Not only do the generated perturbations change the model predictions, but they can also be transferred to other convolutional neural network (CNN) structures. Such a property is called transferability and has been extensively studied. For example, data augmentation-based methods [32, 33] endeavor to create diverse input patterns to enhance representation diversity across different models. Feature disruptive attacks [17, 34, 35] introduce the intermediate loss to change the local activation of image features output from the middle layers of CNNs, enabling the perturbed features to transfer to different models without knowing their structures and parameters. *Text Attack.* Adversarial attacks on natural language processing (NLP) tasks mainly concentrate on word-level and sentence-level perturbations. Word-level perturbations [21, 22, 23, 24] substitute words with synonyms that share similar word embeddings and contextual information. Sentence-level perturbations [25, 36, 37] focus on logical structures of texts through paraphrasing or adding unrelated sentence segments. All of these methods have revealed the adversarial vulnerability of traditional NLP models to some extent. However, adapting them to a multimodal attack setting is still underexplored.

**Multimodal Adversarial Attack Methods**. Multimodal VL models are susceptible to adversarial attacks as perturbations can be added to both modalities. Existing methods mainly explore adversarial robustness on a specific VL task. For the visual question answering task, Fool-VQA [13] is proposed, which iteratively adds pixel-level perturbations on images to achieve the attack. For the image-text retrieval task, CMLA [14] and AACH [38] add perturbations to enlarge Hamming distance between image and text hash codes, which causes wrong image-text matching results. Recently, Co-attack [15] is proposed, which combines image and text perturbations using word substitution attacks to ascertain a direction for guiding the multi-step attack on images. The whole framework is deployed in a white-box setting, where attackers are assumed to have access to the parameters of each downstream task model. Besides, Co-attack is only validated on three tasks, which is not general enough.

## 3  Preliminaries

**VL Pre-trained Model Structure**. Fine-tuning from pre-training has become a unified paradigm in the recent VL model design. Most pre-trained VL models can be divided into *encoder-only* [1, 39, 40, 41] and *encoder-decoder* [4, 5, 6] structures. Two representative model structures are shown in Figure 3. Given an image-text pair $(\mathbf{I}, \mathbf{T})$, both VL models first embed each modality separately. Image tokens are obtained through an image encoder composed of a vision transformer [42] or a Resnet [43] after flattening the grid features into a sequence. Text tokens are generated through

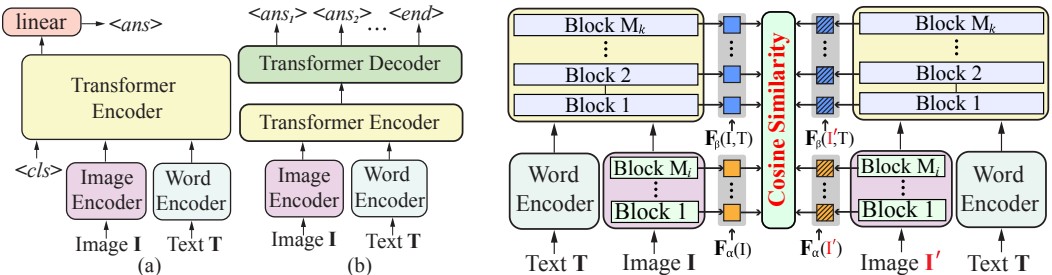

Figure 3: A brief illustration of the encoder-only (a) and encoder-decoder (b) structures.

Figure 4: Block-wise similarity attack. $\mathbf{F}_\alpha$ is the image encoder, and $\mathbf{F}_\beta$ is the Transformer encoder.

a word encoder made up of a tokenizer [44, 45] and a word vector projection. The **encoder-only** model then attends tokens of two modalities and a learnable special token $\langle cls \rangle$ and feeds them into a Transformer encoder. Finally, the output representation from the $\langle cls \rangle$ token is fed into a classification head for the final prediction $\langle ans \rangle$. For the **encoder-decoder** structure, the attached image and text tokens are fed into a Transformer network [46] to generate sequence predictions $[\langle ans_1 \rangle, \langle ans_2 \rangle, \cdots, \langle end \rangle]$ in an auto-regressive manner. The network stops regressing when an end token $\langle end \rangle$ appears. In this work, we deeply explore the adversarial vulnerability of both structures.

**Threat Model**. Let $F$ denote the public pre-trained model and $S$ represent the downstream task model, where $S$ shares most of or the same structure with $F$. As shown in Figure 3, the network structures of both types of public VL pre-trained models are different. The encoder-only model allows modifying the last prediction layer based on the requirement of downstream tasks but keeping the other layers the same as $F$. However, the encoder-decoder model unifies the outputs of different downstream tasks, which leads to $S$ having the same structure as $F$. Note that the downstream tasks will fully fine-tune parameters using their own data, and hence, all the model parameters in $F$ are updated in the fine-tuning stage. Thus, all parameters in $S$ are not accessible.

Given an image-text pair $(\mathbf{I}, \mathbf{T})$, the goal of the downstream task is to predict the labels of the input pair accurately, i.e., $S : (\mathbf{I}, \mathbf{T}) \rightarrow \mathcal{Y}$, where $\mathcal{Y} = \{y_1, \cdots, y_n\}$. For the encoder-only models, the ground truth label is a one-hot vector. For the encoder-decoder models, the ground truth is a sequence that consists of multiple ordered one-hot vectors. Let $\mathbf{y}$ denote the ground truth vector. The goal of the adversarial attack is to generate adversarial examples $(\mathbf{I}', \mathbf{T}')$ using $F$, which can cause an incorrect prediction on $S$. Mathematically, our problem is formulated as follows:

$$\max_{\mathbf{I}', \mathbf{T}'} \mathbb{1}\{S(\mathbf{I}', \mathbf{T}') \neq \mathbf{y}\}, \quad s.t. \ \|\mathbf{I}' - \mathbf{I}\|_\infty < \sigma_i, \ Cos(U_s(\mathbf{T}'), U_s(\mathbf{T})) > \sigma_s, \quad (1)$$

where $\sigma_i$ is the $l_\infty$-norm perturbation strength on the image. $\sigma_s$ is the semantic similarity between the original and perturbed texts, which constrains the semantic consistency after perturbation. The semantic similarity is measured by the cosine similarity $Cos(\cdot, \cdot)$ between the sentence embedding $U_s(\mathbf{T}')$ and $U_s(\mathbf{T})$ using the Universal Sentence Encoder $U_s$ [47].

## 4 Methodology

As shown in Figure 2, the proposed VLATTACK generates adversarial samples from two steps, constrained by perturbation budget parameters $\sigma_i$ and $\sigma_s$. The first step attacks every single modality independently to avoid unnecessary modifications. Samples that fail at the first step will be fed into our multimodal level, where we adopt a cross-search attack strategy to iteratively refine both image and text perturbations at the same time. Next, we will introduce the details of VLATTACK from the single-modal level to the multimodal level.

### 4.1 Single-modal Attacks: From Image to Text

VLATTACK attempts to generate adversarial samples on every single modality in the first step. Compared to the discrete words in $\mathbf{T}$, the continuous values-based image $\mathbf{I}$ is more vulnerable to attack by using gradient information[3]. Thus, VLATTACK starts by attacking the image $\mathbf{I}$.

---

[3]This motivation aligns with our experimental results as shown in Table 1.

**Image-Attack.** In the black-box setting, the parameters of the fine-tuned model $S$ are unknown. However, the pre-trained model $F$ is accessible. Intuitively, if the feature representations learned by the pre-trained model $F$ from the clean input $\mathbf{I}$ and the perturbed input $\mathbf{I}'$ are significantly different, such a perturbation may *transfer* to fine-tuned models to change the predictions of downstream tasks. As shown in Figure 3, the image features can be obtained from both the image encoder and the Transformer encoder, even from different layers or blocks. To fully leverage the specific characteristics of the pre-trained model structure, we propose the **block-wise similarity attack** (BSA) to corrupt universal contextualized representations.

As shown in Figure 4, BSA perturbs images by maximizing the block-wise distances between the intermediate representations in the image encoder $\mathbf{F}_\alpha$ and Transformer encoder $\mathbf{F}_\beta$ of the pre-trained model $F$. Mathematically, we define the loss function of BSA as follows:

$$
\mathcal{L} = \underbrace{\sum_{i=1}^{M_i}\sum_{j=1}^{M_j^i} Cos(\mathbf{F}_\alpha^{i,j}(\mathbf{I}),\ \mathbf{F}_\alpha^{i,j}(\mathbf{I}'))}_{\text{Image Encoder}} + \underbrace{\sum_{k=1}^{M_k}\sum_{t=1}^{M_t^k} Cos(\mathbf{F}_\beta^{k,t}(\mathbf{I},\mathbf{T}),\ \mathbf{F}_\beta^{k,t}(\mathbf{I}',\mathbf{T}))}_{\text{Transformer Encoder}}, \qquad (2)
$$

where $M_i$ is the number of blocks in the image encoder, and $M_j^i$ is the number of flattened image feature embeddings generated in the $i$-th block[4]. Similarly, $M_k$ is the number of blocks in the Transformer encoder, and $M_t^k$ is the number of image token features generated in the $k$-th block. $\mathbf{F}_\alpha^{i,j}$ is the $j$-th feature vector obtained in the $i$-th layer of the image encoder, and $\mathbf{F}_\beta^{k,t}$ is the $t$-th feature vector obtained in the $k$-th layer of the Transformer encoder. The image encoder only takes a single image $\mathbf{I}$ or $\mathbf{I}'$ as the input, but the Transformer encoder will use both image and text as the input. We adopt the cosine similarity to calculate the distances between perturbed and benign features as token representations attended with each other in the inner product space [46]. Note that BSA *does not rely on the information from the decision boundary, and thus, it can be easily adapted to different task settings by disrupting the benign representations.*

In the image attack step, we generate an adversarial image candidate through project gradient decent optimization [30] with $N_s$ iterations, where $N_s < N$ and $N$ is the maximum number of iterations on the image attack. The remaining $N - N_s$ attack step budgets will be used in the multimodal attack in Section 4.2. If $(\mathbf{I}', \mathbf{T})$ is an adversarial sample generated by BSA, then VLATTACK will stop. Otherwise, VLATTACK moves to attack the text modality.

**Text-Attack.** In some VL tasks, the number of tokens in the text is quite small. For example, the average length of the text in the VQAv2 [48] and RefCOCO [49] datasets is 6.21 and 3.57, respectively[5]. Moreover, some of them are nonsense words, which makes it unnecessary to design a new approach for attacking the text modality. Furthermore, existing text attack approaches such as BERT-Attack [21] are powerful for generating adversarial samples for texts. Therefore, we directly apply BERT-Attack to generate text perturbations. To avoid unnecessary modifications, we use the clean image $\mathbf{I}$ as the input instead of the generation perturbed image $\mathbf{I}'$.

Specifically, an adversarial candidate $\mathbf{T}_i'$ produced by BERT-Attack is firstly fed into a universal sentence encoder $U_s$ with the benign text $\mathbf{T}$ to test the semantic similarity. $\mathbf{T}_i'$ will be removed if the cosine similarity $\gamma_i$ between $\mathbf{T}_i'$ and $\mathbf{T}$ is smaller than the threshold $\sigma_s$, i.e., $\gamma_i = Cos(U_s(\mathbf{T}_i'), U_s(\mathbf{T})) < \sigma_s$. Otherwise, we put $\mathbf{T}_i'$ and the benign image $\mathbf{I}$ into the fine-tuned model $S$ to detect whether the new pair $(\mathbf{I}, \mathbf{T}_i')$ perturbs the original prediction $\mathbf{y}$. During the text attack, we create a list $\mathcal{T}$ to store all perturbed samples $\{\mathbf{T}_i'\}$ and the corresponding cosine similarity values $\{\gamma_i\}$. If any sample in $\mathcal{T}$ successfully changes the prediction of the downstream task, then VLATTACK will stop. Otherwise, VLATTACK will carry the perturbed image $\mathbf{I}'$ from the image attack and the perturbed text candidate list $\mathcal{T}$ from the text attack to the multimodal attack level. Note that BERT-Attack is based on synonym word substitution. Even for a short text, the number of adversarial examples can still be large. In other words, the length of $\mathcal{T}$ may be large. The whole single-modal attack process is shown in Algorithm 1 (lines 1-15).

---

[4]If image encoder is a ResNet module, the output image feature from $i$-th block has a shape of $(H_i, W_i, C_i)$, where $H_i$, $W_i$ and $C_i$ denote the height, width and number of channels, respectively. We then collapse the spatial dimensions of the feature and obtain $M_j^i = H_i W_i$.

[5]We obtained these statistics by investigating the validation datasets.

---

**Algorithm 1** VLATTACK

---

**Input:** A pre-trained model $F$, a fine-tuned model $S$, a clean image-text pair $(\mathbf{I}, \mathbf{T})$ and its prediction $y$ on the $S$ , and the Gaussian distribution $\mathcal{U}$;

**Parameters:** Perturbation budget $\sigma_i$ on $\mathbf{I}$, $\sigma_s$ on $\mathbf{T}$. Iteration number $N$ and $N_s$.

1: //Single-modal Attacks: From Image to Text (Section 4.1)
2: **Initialize** $\mathbf{I}' = \mathbf{I} + \delta$, $\delta \in \mathcal{U}(0, 1)$, $\mathcal{T} =$
3: // Image attack by updating $\mathbf{I}'$ using Eq. (2) for $N_s$ steps
4: $\mathbf{I}' = \text{BSA}(\mathcal{L}, \mathbf{I}', \mathbf{T}, N_s, \sigma_i, F)$
5: **if** $S(\mathbf{I}', \mathbf{T}) \neq y$ **then return** $(\mathbf{I}', \mathbf{T})$
6: **else**
7:      // Text attack by applying BERT-attack
8:      **for** pertubed text $\mathbf{T}'_i$ in BERT-attack **do**
9:          **if** $\gamma_i = Cos(U_s(\mathbf{T}'_i), U_s(\mathbf{T})) > \sigma_s$ **then**
10:              Add the pair $(\mathbf{T}'_i, \gamma_i)$ into $\mathcal{T}$;
11:              **if** $S(\mathbf{I}, \mathbf{T}'_i) \neq y$ **then return** $(\mathbf{I}, \mathbf{T}'_i)$
12:              **end if**
13:          **end if**
14:      **end for**
15: **end if**
16: // Multimodal Attack (Section 4.2)
17: Rank $\mathcal{T}$ according to similarity scores $\{\gamma_i\}$ and get top-$K$ samples $\{\hat{\mathbf{T}}'_1, \cdots, \hat{\mathbf{T}}'_K\}$ according to Eq. (3);
18: **for** $k = 1, \cdots, K$ **do**
19:      **if** $S(\mathbf{I}'_k, \mathbf{T}'_k) \neq y$ **then return** $(\mathbf{I}'_k, \mathbf{T}'_k)$
20:      **end if**
21:      Replace $(\mathbf{I}'_k, \hat{\mathbf{T}}'_k)$ with $(\mathbf{I}', \mathbf{T})$ in Eq. (2);
22:      $\mathbf{I}'_{k+1} = \text{BSA}(\mathcal{L}, \mathbf{I}'_k, \hat{\mathbf{T}}'_k, N_k, \sigma_i, F)$
23:      **if** $S(\mathbf{I}'_{k+1}, \mathbf{T}'_k) \neq y$ **then return** $(\mathbf{I}'_{k+1}, \mathbf{T}'_k)$
24:      **end if**
25: **end for**
26: **return None**

---

## 4.2 Multimodal Attack

In many cases, only perturbing images or texts is hard to succeed, as a single-modality perturbation is insufficient to break down the image-text correlation. To solve this problem, we propose a new attack strategy to disrupt dynamic mutual connections based on perturbations from different modalities.

**Attack Steps**. Since the maximum number of image attack steps $N$ is predefined, it is impossible to test all the perturbed image-text pairs $(\mathbf{I}', \mathbf{T}'_i)$ using the remaining budget $N - N_s$ if the length of $\mathcal{T}$ (i.e., $|\mathcal{T}|$) is very large. Thus, we need to rank the perturbed text samples $\{\mathbf{T}'_i\}$ according to their corresponding cosine similarity values $\{\gamma_i\}$ in a descending order to make the adversarial sample keep the high semantic similarity with the original text $\mathbf{T}$. Let $K$ denote the number of attack steps in the multimodal attack, and we have:

$$K = \begin{cases} N - N_s, & \text{if } |\mathcal{T}| > N - N_s; \\ |\mathcal{T}|, & \text{if } |\mathcal{T}| \leqslant N - N_s. \end{cases} \tag{3}$$

**Iterative Cross-Search Attack**. A naive way to conduct the multimodal attack is to test each perturbed image-text pair $(\mathbf{I}', \hat{\mathbf{T}}'_k)$ $(k = 1, \cdots, K)$ by querying the black-box fine-tuned model $S$, where $\hat{\mathbf{T}}'_i$ is the $i$-th text perturbation in the ranked list $\mathcal{T}$. However, this simple approach ignores learning mutual connections between the perturbed text and image. To solve this issue, we propose a new *iterative cross-search attack* (ICSA) strategy. In ICSA, VLATTACK will dynamically update the image perturbation under the guidance of the text perturbation.

Specifically, in each multimodal attack step, ICSA first determines the number of image attack steps. Since there are $K$ adversarial text candidates that will be equally used in ICSA, the iteration number of image attacks that will be allocated to each text sample is $N_k = \lfloor \frac{N - N_s}{K} \rfloor$. ICSA will take the $k$-th pair $(\mathbf{I}'_k, \hat{\mathbf{T}}'_k)$ as the input to generate the new image perturbation $\mathbf{I}'_{k+1}$ by optimizing the block-wise attack loss in Eq. (2), where $\mathbf{I}'_1$ is the output from the image attack in the single-modal level, i.e., $\mathbf{I}'$. Such a process is repeated until an adversarial sample $(\mathbf{I}'_j, \hat{\mathbf{T}}'_j)$ is found, or all the $K$ perturbed texts

Table 1: Comparison of VLATTACK with baselines on ViLT, Unitab, and OFA for different tasks, respectively. All results are displayed by ASR (%). B&A means the BERT-Attack approach.

| Pre-trained Model | Task | Dataset | Image Only | | | | Text Only | | multimodality | |
|---|---|---|---|---|---|---|---|---|---|---|
| | | | DR | SSP | FDA | BSA | B&A | R&R | Co-Attack | VLATTACK |
| **ViLT** | VQA | VQAv2 | 23.89 | 50.36 | 29.27 | 65.20 | 17.24 | 8.69 | 35.13 | **78.05** |
| | VR | NLVR2 | 21.58 | 35.13 | 22.60 | 52.17 | 32.18 | 24.82 | 42.04 | **66.65** |
| **BLIP** | VQA | VQAv2 | 7.04 | 11.84 | 7.12 | 25.04 | 21.04 | 2.94 | 14.24 | **48.78** |
| | VR | NLVR2 | 6.66 | 6.88 | 10.22 | 27.16 | 33.08 | 16.92 | 8.70 | **52.66** |
| **Unitab** | VQA | VQAv2 | 22.88 | 33.67 | 41.80 | 48.40 | 14.20 | 5.48 | 33.87 | **62.20** |
| | REC | RefCOCO | 21.32 | 64.56 | 75.24 | 89.70 | 13.68 | 8.75 | 56.48 | **93.52** |
| | REC | RefCOCO+ | 26.30 | 69.60 | 76.21 | 90.96 | 6.40 | 2.46 | 68.69 | **93.40** |
| | REC | RefCOCOg | 26.39 | 69.26 | 78.64 | 91.31 | 22.03 | 18.52 | 65.50 | **95.61** |
| **OFA** | VQA | VQAv2 | 25.06 | 33.88 | 40.02 | 54.05 | 10.22 | 2.34 | 51.16 | **78.82** |
| | VE | SNLI-VE | 13.71 | 15.11 | 20.90 | 29.19 | 10.51 | 4.92 | 18.66 | **41.78** |
| | REC | RefCOCO | 11.60 | 16.00 | 27.06 | 40.82 | 13.15 | 7.64 | 32.04 | **56.62** |
| | REC | RefCOCO+ | 16.58 | 22.28 | 33.26 | 46.44 | 4.66 | 7.04 | 45.28 | **58.14** |
| | REC | RefCOCOg | 16.39 | 24.80 | 33.22 | 54.63 | 19.23 | 15.13 | 30.53 | **73.30** |

are visited. When $\mathcal{T} = \emptyset$, VLATTACK will degenerate to the block-wise similarity attack (BSA) and iteratively updates $\mathbf{I}'$ for $N - N_s$ steps. Finally, VLATTACK generates adversarial samples based on either single-modality or cross-modality attacks. The overall scheme of VLATTACK is summarized in Algorithm 1.

## 5 Experiments

### 5.1 Experiment Setup

**Pre-trained VL Models and Tasks.** Experiments are conducted on five pre-trained models For the **encoder-only** model, we adopt ViLT [1] and BLIP [28] for two downstream tasks, including the visual question answering (VQA) task on the VQAv2 dataset [48] and the visual reasoning (VR) task on the NLVR2 dataset [50]. For the **encoder-decoder** structure, we adopt Unitab [4] and OFA [5]. For Unitab, evaluations are made on the VQAv2 dataset for the VQA task and on RefCOCO, RefCOCO+, and RefCOCOg datasets [49] for the Referring Expression Comprehension (REC) task that can be viewed as the task of bounding box generation. For OFA, we implement experiments on the same tasks as Unitab and add the SNLI-VE dataset [51] for the visual entailment (VE) task. The specific structures of these models are detailed in Appendix A. To verify the overall generality, we evaluate the uni-modal tasks on OFA [5] using MSCOCO [52] for the image captioning task and ImageNet-1K [53] for the image classification task. We also evaluate CLIP [29] on the image classification task on the SVHN [54] dataset. Note that all evaluation tasks have publicly available pre-trained and fine-tuned models, which provide more robust reproducibility. The details of each dataset and implementation can be found in Appendix B.

**Baselines**. We compare VLATTACK with adversarial attacks on different modalities. For attacks on the **image** modality, we take DR [17], SSP [35], and FDA [34] as baselines. These methods are designed to perturb image features only and can be directly adapted to our problem. Other methods [18, 19, 30, 32, 33, 55] either fully rely on the output from classifiers or combine feature perturbation with classification loss [20, 56, 57]. These methods can not be applied in our problem setting since the pre-trained and fine-tuned models usually share different prediction heads and are trained on different tasks. For attacks on the **text** modality, we take BERT-Attack (B&A) [21] and R&R [24] as baselines. VLATTACK also compares with Co-Attack [15], which is the only **multimodal** attack scheme that adds adversarial perturbations to both modalities.

### 5.2 Results on Multimodal Tasks

In this section, we conduct experiments on four pre-trained VL models and use **Attack Success Rate** (ASR) to evaluate the performance on four multimodal tasks, including VQA, VR, REC, and VE. The higher the ASR, the better the performance. Results are illustrated in Table 1, where the results of our proposed BSA and VLATTACK are highlighted. We can observe that the proposed VLAT-TACK significantly outperforms all baselines. Compared with the best baseline on each dataset,

Table 2: Evaluation of the Uni-modal tasks on OFA. We highlight the prediction score reported by the original OFA paper with $*$.

| Dataset | MSCOCO | | | | ImageNet-1K |
|---|---|---|---|---|---|
| Metric | BLEU@4 ($\downarrow$) | METEOR ($\downarrow$) | CIDEr ($\downarrow$) | SPICE ($\downarrow$) | ASR($\uparrow$) |
| OFA$*$ | 42.81 | 31.30 | 145.43 | 25.37 | - |
| DR | 30.26 | 24.47 | 95.52 | 17.89 | 10.43 |
| SSP | 10.99 | 12.52 | 23.54 | 5.67 | 19.44 |
| FDA | 17.77 | 17.92 | 55.75 | 11.36 | 12.31 |
| BSA (Ours) | 3.04 | 8.08 | 2.16 | 1.50 | 41.35 |

Table 3: CLIP model evaluation on SVHN.

| Dataset | SVHN | |
|---|---|---|
| Model | CLIP-ViT/16 | CLIP-RN50 |
| DR | 3.32 | 71.62 |
| SSP | 6.36 | 84.26 |
| FDA | 6.20 | 83.52 |
| BSA (Ours) | 15.74 | 84.98 |

Table 4: MTurK evaluation on the ViLT model using the VQAv2 dataset. Results are obtained on 650 samples.

| Method | SSP | VLATTACK |
|---|---|---|
| Definitely Correct ($\downarrow$) | 413 | 307 |
| Not Sure ($\uparrow$) | 58 | 56 |
| Definitely Wrong ($\uparrow$) | 179 | 287 |

VLATTACK achieves an average gain of **29.61**% on ViLT, **24.20**% on BLIP, **16.48**% on Unitab and **30.84**% on OFA, respectively. When only adding **image** perturbations, our proposed BSA outperforms the best baseline by an average of 15.94% on ViLT and on BLIP. BSA also achieves an average gain of 12.12% and 14.13% on Unitab and OFA, respectively. As mentioned before, **text** data in multimodal tasks are composed of phrases or only a few words. Consequently, text attack performs poorly on most multimodal datasets. These high ASR values have revealed significant security concerns regarding the adversarial robustness of pre-trained VL models.

### 5.3 Results on Uni-Modal Tasks

For the results on uni-modal tasks, we first evaluate OFA on image captioning and classification tasks on MSCOCO and ImageNet-1K, respectively. *Because these tasks accept a fixed text prompt for prediction, we only perturb image* $\mathbf{I}$ *and compare different image attack methods.* For the captioning task, we choose four commonly-used metrics, including BLEU@4, METEOR, CIDEr, and SPICE, the same as those used in OFA [5]. The lower, the better. We report ASR on the image classification task. The higher, the better. The experimental results are illustrated in Table 2. In general, BSA achieves the best performance on both tasks in terms of all metrics. For example, BSA significantly reduces the CIDEr score from 145.43 to **2.16**. Compared with the best baseline SSP with 23.54, BSA reduces this score over **10 times**. For the image classification task, BSA also achieves the best performance with **41.35**% ASR. The superior performance demonstrates that the VLATTACK can also generalize to computer vision tasks when only using BSA. Note that we do not evaluate text-to-image generation tasks because our attack framework produces the same results as BERT-Attack with text-only inputs.

We also evaluate CLIP on the image classification task on the SVHN dataset. Concretely, we use the image encoder of CLIP as the pre-trained model and then fine-tune on the SVHN dataset after adding a linear classification head. For the choices of image encoder of CLIP, we adopt ViT-B/16 and ResNet-50, denoted by CLIP-ViT/16 and CLIP-RN50. We test the attack performance using 5K correctly predicted samples. All results are illustrated in the following Table 3. Since the task only accepts images as input, we compare our BSA with other image attack baselines. As shown in the table, our proposed BSA still maintains the best ASR using different image encoder structures, clearly demonstrating its effectiveness.

### 5.4 Systematic Validation with Crowdsourcing

We also conduct a human evaluation study to comprehensively verify the soundness of our proposed method. The experiment is developed on the results output from the ViLT model on the VQAv2 dataset using Amazon Mechanical Turk (MTurk). The baseline that we choose is SSP since it outperforms others in our experiments, as shown in Table 1. Specifically, we randomly sampled 650 examples and stored the generated answers after the attack. To validate the performance of both approaches, we send the original image-question pairs and the corresponding generated answer to

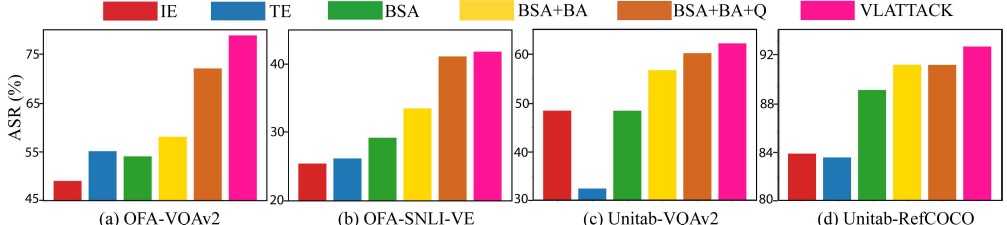

Figure 5: Ablation analysis of different components in VLATTACK. We show the results of VQAv2 (a) and SNLI-VE (b) on OFA, and VQAv2 (c) and RefCOCO (d) on Unitab.

the MTurk system. We provide three choice candidates to workers, including "Definitely Correct", "Not Sure", and "Definitely Wrong". A successful attack means that the worker will label "Definitely Wrong" to the pair. To make the annotations more accurate and reliable, each pair is annotated by three workers, and we report the majority choice as the final human evaluation result. The statistics of human evaluation results are shown in Table 4. We can observe that the proposed VLAttack still significantly outperforms the strongest baseline SSP, thus demonstrating its robustness and effectiveness from a human perceptual perspective.

## 5.5 Model Design Analysis

**Ablation Study.** In our model design, we propose a new block-wise similarity attack (BSA) for attacking the image modality and an interactive cross-search attack (ICSA) for attacking image-text modalities together. We use the following methods to evaluate the effectiveness of each component. The results are shown in Figure 5. "IE"/"TE" means that we only use the image/Transformer encoder when calculating the loss in Eq. (2). "BSA" uses both encoders. We set iteration $N_s = 40$ for a fair comparison. Next, "BSA+BA" means after attacking images using BSA, we attack texts using BERT-Attack (Algorithm 1 Lines 1-15). "BSA+BA+Q" denotes replacing ICSA with a simple query strategy by querying the black-box task with each pair $(\mathbf{I}', \mathbf{T}'_i)$, where $\mathbf{T}'_i$ comes from the perturbed text candidate list $\mathcal{T}$. We can observe that for image attacks, IE and TE play an important role in different tasks, but in general, BSA (= IE + TE) outperforms them. Adding the text attack, BSA+BA performs better than BSA. This comparison result demonstrates that both modalities are vulnerable. BSA+BA+Q performs better than BSA+BA but worse than VLATTACK, which confirms the necessity and reasonableness of conducting the interactive cross-search attack. More results on other datasets are shown in Appendix C.

**Parameter Sensitivity Analysis** We discuss the effect of different iteration numbers of $N$ and $N_s$ in VLATTACK. All experiments are conducted on the VQAv2 dataset and the ViLT model. The total iteration number $N$ is set from 10 to 80, $N_s$ is set to $\frac{N}{2}$. As depicted in Figure 6(a), the ASR performance is dramatically improved by increasing $N$ from 10 to 20 steps and then achieves the best result when $N = 40$. We next investigate the impact of different initial iteration numbers $N_s$. We test $N_s$ from 5 to 40, but the total iteration number $N$ is fixed to 40. As shown in Figure 6(b), the ASR score reaches the summit when $N_s$ is 5, and it is smoothly decreased by continually enlarging $N_s$. Considering that the smaller initial iteration number $N_s$ increases the ratio of text perturbations, we set $N_s$ as 20 to obtain the best trade-off between attack performance and the naturalness of generated adversarial samples in our experiments.

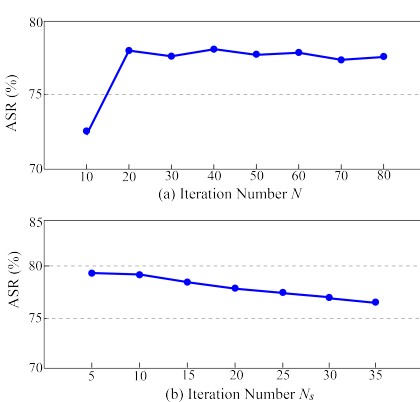

Figure 6: Investigation of iteration number $N$ and $N_s$.

**Block-wise Similarity Attack**. We further visualize the effect of our proposed BSA through attention maps in the fine-tuned downstream model $S$ on the VQAv2 dataset through the ViLT model. As is shown in Figure 7, the attention area is shifted from "*jacket*" to the background, which indicates that the representation used to answer the question extracts information from the unrelated image regions. We also combined the BSA with various optimization methods to verify its generalization ability. The experimental results are presented in Appendix D.

## 5.6 Case Study

We conduct case studies using visualization to show the effectiveness of VLATTACK for multimodal and uni-modal tasks, as displayed in Figure 8. For multimodal tasks in Figure 8(a), the predictions of all displayed samples remain unchanged when attacking a single modality via BSA. However, perturbing both modalities using VLATTACK successfully modifies the predictions. An intriguing finding can be observed in the REC task, where the fine-tuned model $S$ outputs wrong predictions by stretching and deforming the original bounding box predictions. The proposed VLATTACK also shows its power on uni-modal tasks in Figure 8(b). For example, in the image captioning task, the fine-tuned model generates a description of a "*person*" after receiving an adversarial image of a "*horse*", which is entirely unrelated. More case studies are shown in Appendix E.

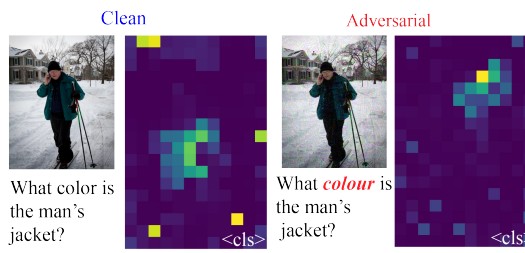

Figure 7: Attention maps of the clean and adversarial samples in the multimodal transformer encoder. Attention maps from the class token $\langle cls \rangle$ to images. Perturbed tokens are displayed in red.

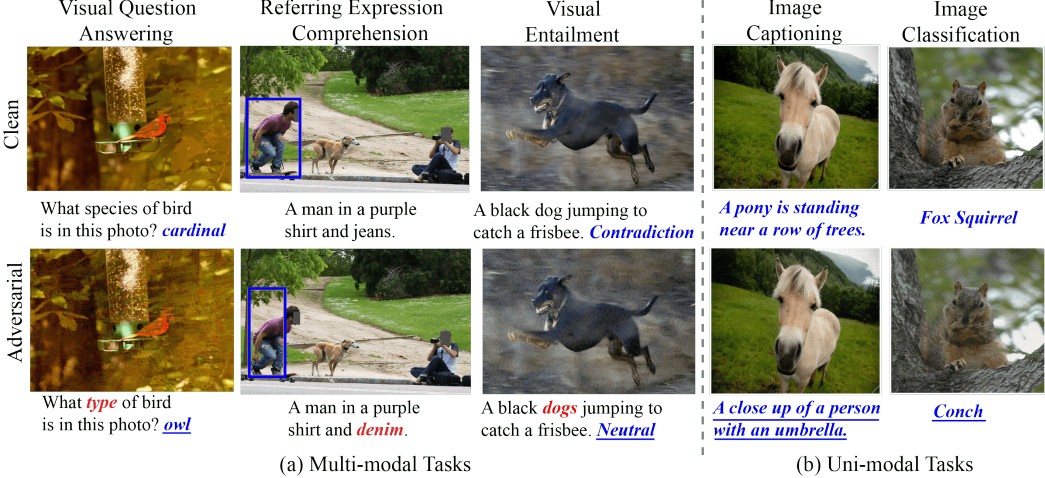

Figure 8: Qualitative results of VLATTACK on (a) multimodal tasks and (b) Uni-modal tasks on OFA. Perturbed word tokens and original predictions are displayed in red and blue, respectively. We show the predictions after the adversarial attack with underline.

## 6 Conclusion

In this paper, we propose a new question, which aims to generate adversarial samples only based on the publicly available pre-trained models to attack fine-tuned models deployed in different VL tasks. Considering the task-specific and model-specific challenges, we proposed VLATTACK, which generates perturbations by exploring different modalities in two levels. The single-modal level attack first perturbs images through a novel algorithm BSA to disrupt the universal image-text representations and then attacks text if the former fails, which avoids unnecessary modifications on both modalities. If both image and text attacks fail, the multimodal attack level adopts an iterative cross-search attack strategy to generate adversarial image-text combinations. By periodically substituting text candidates during the image attack process, the mutual relations between different modal perturbations are sufficiently explored. Experimental results demonstrate the effectiveness of the proposed VLATTACK on attacking multiple VL tasks, which reveals a significant safety concern in realistic scenarios.

**Acknowledgements** This work is partially supported by the National Science Foundation under Grant No. 1951729, 1953813, 2119331, 2212323, and 2238275.

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

# A. Details of VL Models

This section introduces the details of VL models involved in our experiment, including ViLT, BLIP, Unitab, and OFA.

- **ViLT**. We first select ViLT [1] as the **encoder-only** VL model because of its succinct structure and prominent performance on multiple downstream tasks. Given an input image $\mathbf{I} \in \mathbb{R}^{H \times W \times 3}$ and a sentence $\mathbf{T}$, ViLT yields $M$ image tokens using a linear transformation on the flattened image patches, where each token is a 1D vector and $M = \frac{HW}{P^2}$ for a given patch resolution $(P, P)$. Word tokens are encoded through a Byte-Pair Encoder (BPE) [44] and a word-vector linear projection. Then, tokens of two modalities and a special learnable token $\langle cls \rangle$ are concatenated. By attending visual and text tokens and a special token $\langle cls \rangle$ in a Transformer encoder with twelve layers, the output feature from the $\langle cls \rangle$ token is fed into a task-specific classification head for the final output. Taking the VQA task as an example, the VQA classifier adopts a linear layer to output a vector with $H_s$ elements, where $H_s$ is the number of all possible choices in the closed answer set of the VQA task. The final output is obtained through the element with the highest response in the vector.

- **BLIP**. The BLIP model also adopts an **encoder-only** structure. Specifically, BLIP first encodes an image through a twelve-layer vision transformer ViT-B/16 [42], and generates word token embeddings through the BPE and a linear layer. Next, the image features and word token embeddings are then fed into a multimodal encoder. The structure of the multimodal encoder is the same as a twelve-layer transformer decoder, where each layer contains a self-attention module, a cross-attention module and a feed-forward module. In each layer, the multimodal encoder first accepts the word token embeddings as input and attends to them through the self-attention module. Then, the updated embedding will fuse with image features through the cross-attention module. Finally, the output from the multimodal encoder will be fed into different projection heads for downstream tasks, just like the ViLT model.

- **Unitab.** Unitab adopts an **encoder-decoder** framework. It first embeds text $\mathbf{T}$ via RoBERT$_a$ [45] and flats features after encoding image $\mathbf{I}$ through ResNet [43]. The attached visual and text token features are then fed into a standard Transformer network [46] with six encoder layers and six decoder layers. Finally, the sequence predictions $[\langle ans_1 \rangle, \langle ans_2 \rangle, \cdots, \langle end \rangle]$ are obtained auto-regressively through a projection head. The network stops regressing when an end token $\langle end \rangle$ appears. For different tasks, the output tokens may come from different pre-defined vocabularies. Given the REC task as an example, four tokens $[(\langle loc\ x_1 \rangle, \langle loc\ x_2 \rangle), (\langle loc\ x_3 \rangle, \langle loc\ x_4 \rangle)]$ will be selected from the location vocabulary, which forms the coordinate of a bounding box. As a result, these models can handle both text and grounding tasks.

- **OFA.** OFA also adopts an **encoder-decoder** structure. Different from Unitab, it adopts the BPE to encode text and extends the linguistic vocabulary by adding image quantization tokens [58] $\langle img \rangle$ for synthesis tasks. ***Note that the main difference between OFA and Unitab lies in their pre-training and fine-tuning strategies rather than the model structure***. For example, in the pre-training process, Unitab focuses on learning alignments between predicted words and boxes through grounding tasks, while OFA captures more general representations through multi-task joint training that includes both single-modal and multimodal tasks. Overall, OFA outperforms Unitab in terms of performance improvement.

# B. Dataset and Implementation

### B.1 Tasks and Datasets

To verify the generalization ability of our proposed VLATTACK, we evaluate a wide array of popular vision language tasks summarized in Table 5. Specifically, the selected tasks span from text understanding (visual reasoning, visual entailment, visual question answering) to image understanding (image classification, captioning) and localization (referring expression comprehension).

For each dataset, we sample 5K **correctly predicted samples** in the corresponding validation dataset to evaluate the ASR performance. All validation datasets follow the same split settings as adopted in the respective attack models. Because VQA is a multiclass classification task, we select a correct

Table 5: An illustration of all datasets and tasks evaluated in our paper.

| Datasets | Task | Task description | Attack Model | | | Attack Modality | |
|---|---|---|---|---|---|---|---|
| | | | OFA | Unitab | ViLT | Image | Text |
| VQAv2 | VQA | Scene Understanding QA | ✓ | ✓ | ✓ | ✓ | ✓ |
| SNLI-VE | VE | VL Entailment | ✓ | | | ✓ | ✓ |
| RefCOCO | REC | Bounding Box Localization | ✓ | ✓ | | ✓ | ✓ |
| RefCOCOg | REC | Bounding Box Localization | ✓ | ✓ | | ✓ | ✓ |
| RefCOCO+ | REC | Bounding Box Localization | ✓ | ✓ | | ✓ | ✓ |
| NLVR2 | VR | Image-Text Pairs Matching | | | ✓ | ✓ | ✓ |
| MSCOCO | Captioning | Image Captioning | ✓ | | | ✓ | |
| ImageNet-1K | Classification | Object Classification | ✓ | | | ✓ | |
| SVHN | Classification | Digit Classification | ✓ | | | ✓ | |

prediction only if the prediction result is the same as the label with the highest VQA score[6], and regard the label as the ground truth in Eq. (1). In the REC task, a correct prediction is considered when the Intersection-over-Union (IoU) score between the predicted and ground truth bounding box is larger than 0.5. We adopt the same IoU threshold as in Unitab [4] and OFA [5].

### B.2 Implementation Details

For the perturbation parameters of images, we follow the setting in the common transferable image attacks [18, 19] and set the maximum perturbation $\sigma_i$ of each pixel to 16/255 on all tasks except REC. Considering that even a single coordinate change can affect the final grounding results to a great extent, the $\sigma_i$ of the REC task is 4/255 to better highlight the ASR differences among distinct methods. The total iteration number $N$ and step size are set to 40 and 0.01 by following the projected gradient decent method [30], and $N_s$ is 20. For the perturbation on the text, the semantic similarity constraint $\sigma_s$ is set to 0.95, and the number of maximum modified words is set to 1 by following the previous text-attack work [15, 24] to ensure the semantic consistency and imperceptibility. All experiments are conducted on a single GTX A6000 GPU.

## C. More Ablation Results

In Section 5.4, we conduct an ablation study to show the effectiveness of each module in our model design on VQA, VE, and REC tasks. Here, we conduct additional ablation experiments for the remaining tasks, including visual reasoning, image captioning, and image classification.

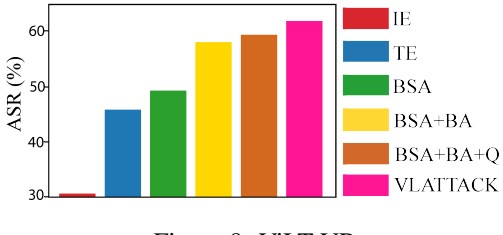

Figure 9: ViLT-VR.

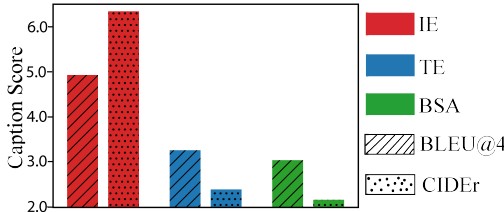

Figure 10: OFA-captioning.

Figure 9 shows the results of the ablation study on the VR task using the ViLT model. We can observe that only using the image encoder results in significantly low ASR. However, by combining it with the Transformer encoder (TE), BSA can achieve a high ASR. These results show the reasonableness of considering two encoders simultaneously when attacking the image modality. The result of BSA+BA demonstrates the usefulness of attacking the text modality. Although BSA+BA+Q outperforms other approaches, its performance is still lower than that of the proposed VLATTACK. This comparison proves that the proposed iterative cross-search attack (ICSA) strategy is effective for the multimodal attack again.

---

[6]The VQA score calculates the percentage of the predicted answer that appears in 10 reference ground truth answers. More details can be found via `https://visualqa.org/evaluation.html`

Figure 10 shows the results of the image captioning task using the OFA model. Because the image captioning task only accepts a fixed text prompt for prediction, we only perturb the image and report the results on IE, TE, and BSA. For this task, we report BLEU@4 and CIDEr scores. **The lower, the better**. We can observe that the proposed BSA outperforms IE and TE, indicating our model design's effectiveness.

Figure 11 shows the results of the image classification task using the OFA model. The experiment is conducted on the ImageNet-1K dataset. Similar to the image captioning task, we only attack images, and compare the results of IE, TE and BSA. The evaluation metric for this task is ASR. **The higher, the better**. We can have the same observations with other ablation studies, where attacking both encoders outperforms attacking a single encoder.

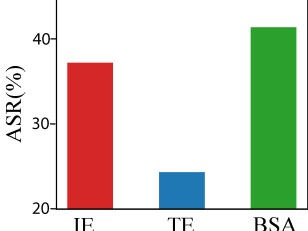
Figure 11: OFA-classification.

Table 6: Combining VLATTACK with different gradient-based image attack schemes.

| Method | ViLT | | Unitab | | | |
|---|---|---|---|---|---|---|
| | VQAv2 | NLVR2 | VQAv2 | RefCOCO | RefCOCO+ | RefCOCOg |
| $\text{BSA}_{MI}$ | 65.40 | 52.32 | 50.38 | 86.00 | 89.20 | 87.39 |
| $\text{VLATTACK}_{MI}$ | **78.77** | **67.16** | **63.02** | **92.46** | **93.10** | **94.34** |
| $\text{BSA}_{DI}$ | 65.94 | 52.30 | 42.74 | 90.30 | 91.56 | 91.00 |
| $\text{VLATTACK}_{DI}$ | **78.07** | **67.50** | **61.22** | **93.98** | **94.04** | **94.76** |

## D. Different Optimization Methods

VLATTACK can be easily adapted to various optimization methods in image attacks. To demonstrate the generalizability of our method, we replace the projected gradient decent [30] in VLATTACK with Momentum Iterative method (MI) [55] and Diverse Input attack (DI) [33] since they have shown better performance than traditional iterative attacks [30, 59]. The replaced methods are denoted by $\text{BSA}_{MI}$, and $\text{VLATTACK}_{MI}$ using MI, $\text{BSA}_{DI}$ and $\text{VLATTACK}_{DI}$ using DI, respectively. Experiments are developed on ViLT and Unitab. Results are shown in Table 6. Using MI and DI optimizations, $\text{BSA}_{MI}$ and $\text{BSA}_{DI}$ still outperform all baselines displayed in Table 1 in the main manuscript. Also, $\text{VLATTACK}_{MI}$ and $\text{VLATTACK}_{DI}$ outperform the image attack method $\text{BSA}_{MI}$ and $\text{BSA}_{DI}$ with an average ASR improvement of 9.70% and 9.29% among all datasets. The gain of performance demonstrates that the proposed VLATTACK can be further improved by combining with stronger gradient-based optimization schemes.

## E. Case Study

### E.1 How does VLATTACK generate adversarial samples?

The proposed VLATTACK aims to attack multimodal VL tasks starting by attacking single modalities. If they are failed, VLATTACK uses the proposed interactive cross-search attack (ICSA) strategy to generate adversarial samples. In this experiment, we display the generated adversarial cases from different attack steps, including the image modality in Figure 12, the text modality in Figure 13, and the multimodal attack in Figure 14.

**Single-modal Attacks (Section 4.1)**. VLATTACK first perturbs the image modality using the proposed BSA and only outputs the adversarial image if the attack is successful (Algorithm 1 lines 1-5). As shown in Figure 12, only attacking the image modality, VLATTACK can generate a successful adversarial sample to fool the downstream task. Then, VLATTACK will stop. Otherwise, it will perturb the text through BERT-Attack (B&A) and use the clean image as the input, which is illustrated in Figure 13 (Algorithm 1, lines 6-15). During the text attack, B&A will generate multiple candidates by replacing the synonyms of a word. Since the length of text sentences is very short in the VL datasets, we only replace one word each time. From Figure 13, we can observe that B&A first replaces "kid" with its synonym "child", but this is not an adversarial sample. B&A then moves to the next word "small" and uses its synonym "cute" as the perturbation. By querying the black-box downstream task model, VLATTACK successes, and the algorithm will stop. **Multimodal Attack**

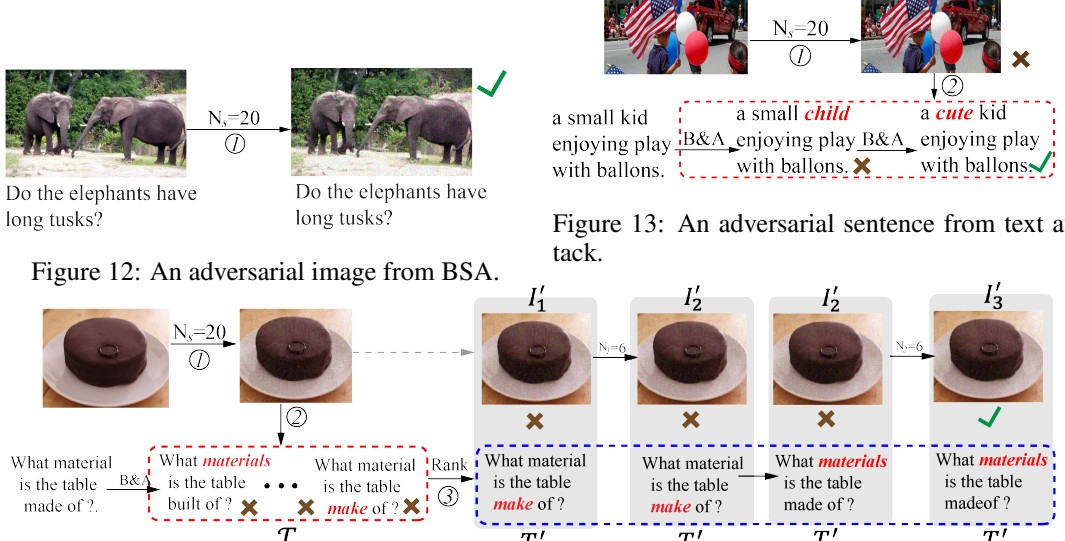

Figure 12: An adversarial image from BSA.

Figure 13: An adversarial sentence from text attack.

Figure 14: An adversarial image-text pair from multimodal attack.

**(Section 4.2)**. If the single-modal attack fails, VLATTACK moves to the multimodal attack by iteratively cross-updating image and text perturbations, where image perturbations are added through BSA, and text perturbations are added according to the semantic similarity. The cross-updating process is repeated until an adversarial image-text pair is found (Algorithm 1, lines 16-24). Figure 14 shows an example. In step 1, VLATTACK fails to attack the image modality and outputs a perturbed image denoted as $\mathbf{I}'_1$. In step 2, VLATTACK also fails to attack the text modality and outputs a list of text perturbations $\mathcal{T}$. VLATTACK has to use the multimodal attack to generate adversarial samples in step 3. It first ranks the text perturbations in $\mathcal{T}$ according to the semantic similarity between the original text and each perturbation. The ranked list is denoted as $\{\hat{\mathbf{T}}'_1, \cdots, \hat{\mathbf{T}}'_K\}$. Then it equally allocates the iteration number of the image attack to generate the image perturbations iteratively. In Figure 14, this number is 6, which means we run BSA with the budget 6 to generate a new image perturbation.

VLATTACK takes the pair $(\mathbf{I}'_1, \hat{\mathbf{T}}'_1)$ as the input to query the black-box downstream model, where $\hat{\mathbf{T}}'_1 =$ "*What material is the table make of?*". If this pair is not an adversarial sample, then the proposed ICSA will adopt BSA to generate the new image perturbation $\mathbf{I}'_2$. The new pair $(\mathbf{I}'_2, \hat{\mathbf{T}}'_1)$ will be checked again. If it is still not an adversarial sample, VLATTACK will use the next text perturbation $\hat{\mathbf{T}}'_2 =$ "*What materials is the table made of?*" and the newly generated image perturbation $\mathbf{I}'_2$ as the input and repeat the previous steps until finding a successful adversarial sample or using up all $K$ text perturbations in $\mathcal{T}$. VLATTACK employs a systematic strategy for adversarial attacks on VL models, sequentially targeting single-modal and multimodal perturbations to achieve successful adversarial attacks.

### E.2 Case Study on Different Tasks

We also provide additional qualitative results from Figure 15 to Figure 20 for experiments on all six tasks. For better visualization, we display the adversarial and clean samples side by side in a single column. By adding pixel and word perturbations, the fidelity of all samples is still preserved, but predictions are dramatically changed. For instance, in the image captioning task of Figure 19, all generated captions show no correlation with the input images. Some texts may even include replacement Unicode characters, such as "\ufffd", resulting in incomplete sentence structures.

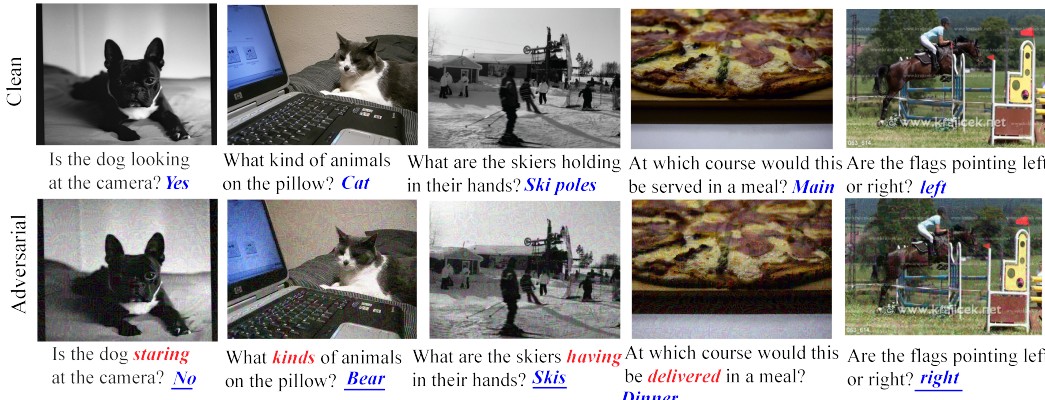

Figure 15: Additional quantitative results on visual question answering (VQA).

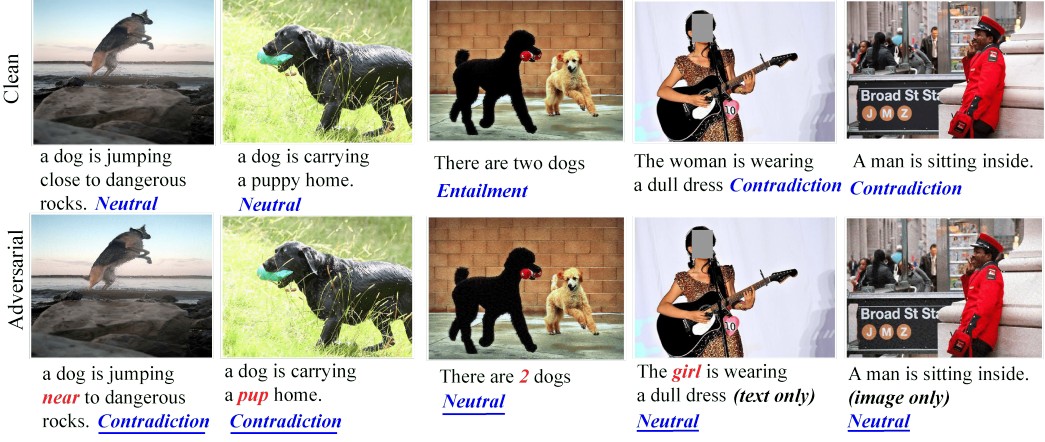

Figure 16: Additional quantitative results on visual entailment (VE).

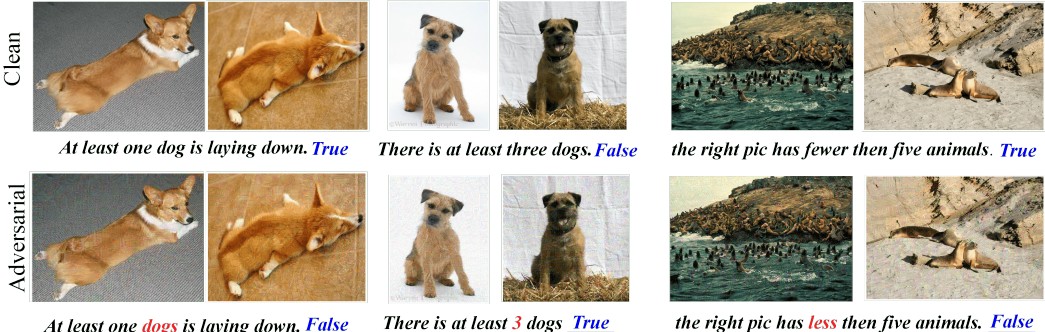

Figure 17: Additional quantitative results on visual reasoning (VR).

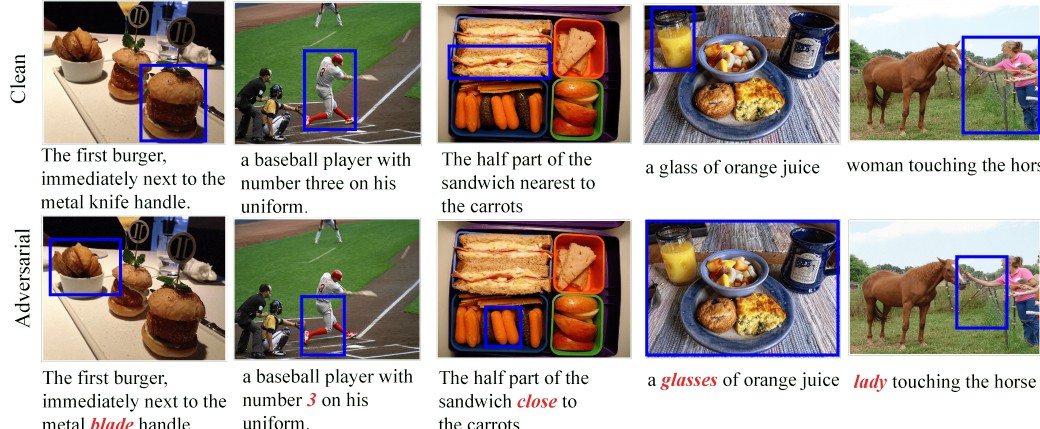

Figure 18: Additional quantitative results on referring expression comprehension (REC).

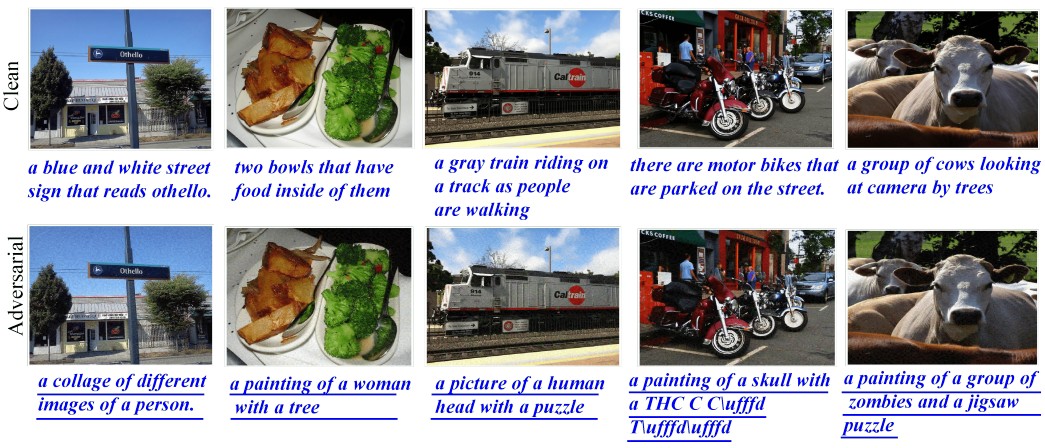

Figure 19: Additional quantitative results on the image captioning task.

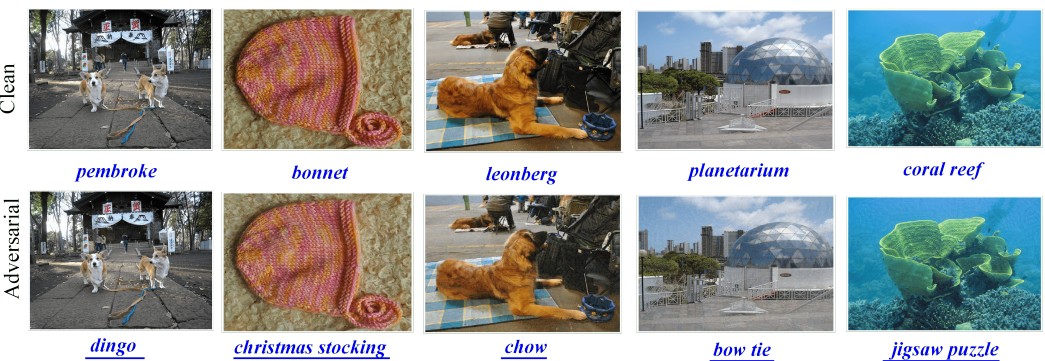

Figure 20: Additional quantitative results on the image classification task.

## F. Discussion: BSA v.s. BadEncoder

We notice that the idea of our proposed BSA is somehow similar to the BadEncoder [60] method, as they all use the cosine similarity as the optimization target. However, the proposed BSA is different from or better than BadEncoder.

(1) BadEncoder only utilizes the final output feature vectors from the whole encoder, ignoring the outputs from the intermediate layers/blocks. However, BSA calculates fine-grained similarity scores. As shown in Eq. 2 of our original paper, we distinguish the outputs from image and Transformer encoders. Such a design can modify the low-level vision features and the high-level cross-modal representations. In our setting, we attack fine-tuned models of a wide diversity of downstream tasks, including but not limited to image classification tasks like Badencoder. The parameters of these task-specific models are fully finetuned on distinct datasets, and the output representations of the encoder significantly change accordingly. Thus, instead of only attacking the output feature from the last layer like BadEncoder, perturbing each intermediate feature representation from each encoder and each block can enhance the attack performance. This statement is also verified in Section 5.5, where the ASR score of BSA is higher than only attacking a single encoder.

(2) The motivation for adopting the cosine distance is different. In BadEncoder, CLIP uses cosine similarity as a loss function to calculate distances for positive/negative image text pairs, which is motivated by the pre-training strategy of CLIP. However, we adopt cosine similarity because the fine-grained token representations attend to each other in the inner product space, which is inspired by the mechanism design of the Transformer structure. Therefore, the proposed BSA method greatly differs from BadEncoder in the above two aspects.

## G. Limitations

The limitations of our work can be summarized from the following two aspects. On the one hand, in our current model design, for the text modality, we directly apply the existing model instead of developing a new one. Therefore, there is no performance improvement on tasks that only accept texts as input, such as text-to-image synthesis. On the other hand, our research problem is formulated by assuming the pre-trained and downstream models share similar structures. The adversarial transferability between different pre-trained and fine-tuned models is worth exploring, which we left to our future work.

## H. Broad Impacts

Our research reveals substantial vulnerabilities in vision-language (VL) pre-trained models, underlining the importance of adversarial robustness cross pre-trained and fine-tuned models. By exposing these vulnerabilities through the VLATTACK strategy, we offer inspiration for developing more robust models. Furthermore, our findings underscore the ethical considerations of using VL models in real-world applications, especially those dealing with sensitive information and big data. Overall, our work emphasizes the necessity of balancing performance and robustness in VL models, with implications extending across computer vision, natural language processing, and broader artificial intelligence applications.

