# OpenReview forum: "VLATTACK: Multimodal Adversarial Attacks on Vision-Language Tasks via Pre-trained Models"
_NeurIPS.cc/2023/Conference — NeurIPS 2023 poster_

### Official Review · Reviewer_zqVz · 2023-06-15

**Soundness:** 2 fair
**Presentation:** 3 good
**Contribution:** 3 good
**Rating:** 5
**Confidence:** 4

**Summary:**

The authors present a new adversarial attack framework, VLATTACK, to evaluate the robustness of large vision-language models (VLMs). The paper introduces a two-step adversarial attack approach. The first step involves attacking each modality (image and text) independently, and the second step employs a cross-search attack strategy to iteratively query the fine-tuned model for perturbation update. The paper conducts experiments on multiple pre-trained models that are fine-tuned for downstream vision tasks.



**Strengths:**

1. The paper is well-written with a clear presentation that demonstrates the proposed framework.

2. The VLATTACK, which is a two-step adversarial attack framework, involves both white-box gradient attack and black-box query attack. This idea is interesting with practical implementation.

3. The evaluation takes account of multiple types of pre-trained models, and the BSA part of the attack overperforms SOTA image-space adversarial baselines.

**Weaknesses:**

1. The black-box setting is questionable. Although the adversary only has black-box access to the downstream fine-tuned model, the first stage of attack has white-box access to widely-used foundations like ViLT. There is a high chance that the victim model shares mutual information (e.g., has knowledge of the same vision-language dataset / has the same model architecture) with the white-box pre-trained model. It will be beneficial if the authors elaborate more on the source of transferability.
2. Section 4.2 (ICSA) claims to be under the black-box setting. However, Lines 228-229 describe the process to update the perturbation by optimizing the attack loss of Eq. (2). Authors should write clearly how the gradient is obtained, and what model is white-box during this step.
3. Section 4.1 Text-Attack part uses clean image $\mathbf{I}$ instead of generated adversary $\mathbf{I'}$ for text optimization. Authors claim that this is to avoid unnecessary modification. However, using $\mathbf{I'}$ can possibly ease the difficulty of textual search and increase the success rate of adversarial textual optimization. Moreover, the authors should argue clearly what the unnecessary modification stands for.
4. The success of the attack will require a sufficient number of queries to generate effective adversaries. It will be helpful if authors evaluate the query cost (e.g., time of inference/API call), and discuss attack effectiveness by the number of queries.

**Questions:**

1. Fig. 7 second column demonstrates that the model comprehension shifts from *A man in a purple shirt and **jeans*** to *A man in a purple shirt and **denim***. However, denim is the material pattern for the jeans shown in the image. The model prediction is still consistent with the image. It will be beneficial if the authors state clearly what this example stands for.

2. Given the setting of this paper that we have (1) publicly available pre-trained models and (2) the clean dataset for future perturbation, I wonder if it is possible to first fine-tune the model on the clean dataset for the downstream task and then conduct the transfer-based attack? It will be beneficial for this paper to argue the infeasibility of such approaches.

3. To summarize, with a fair paper presentation but several concerns stated in the weakness/question sections, I will rate it as a borderline accept at this stage. However, I look forward to the authors' response and I will consider revising the rating based on the soundness of the responses.


**Limitations:**

Yes, the authors have stated the potential limitations of the work in the supplementary materials.

---

> ### Author Rebuttal · Authors · 2023-08-09
>
> Thank you for the valuable review. \
> `>>> W1` ***Elaborate more on the source of transferability***\
> `>>> A`:  We agree that the pre-trained and fine-tuned models share almost the same architecture in the current setting. Such mutual information should help to improve the attack success rate.
>
> As suggested by the reviewer, we further design an experiment of __*transfer attacks on different structures*__. Based on the VQA task, we first generate adversarial samples on the pretrained ViLT and use them to attack the fine-tuned ALBEF model. The two models have different architectures. The results are shown in the table.
>
> |Method| SSP  | Co-Attack | VLAttack |
>  |-| - | - |-|
> |ASR|12.78  | 13.04 |__15.28__ |
>
> Although VLAttack is not explicitly designed for this scenario, it achieves the best ASR. Also, the relatively low ASR suggests potential space to explore this setting in the future.
>
> `>>> W2` ***How is the gradient obtained?***\
>  `>>> A`: Thanks for the advice. We first feed the output from the previous step \$\boldsymbol{I}_{k}^{'}\$ to the pre-trained model $F$. Then we compute the BSA loss by replacing \$(\boldsymbol{I}_k^{'},\boldsymbol{T}_k^{'})\$ with \$(\boldsymbol{I}^{'},\boldsymbol{T})\$ in Eq.(2) of our paper. The gradients are then derived on the $F$ and will be clipped to produce the new perturbed image of the $k+1$ step. So the pre-trained model $F$ is the only white-box model during this process.
>
> `>>> W3` ***Using $\boldsymbol{I}^{'}$ in the text-attack stage may increase ASR, and what does the '' unnecessary modification '' mean?***\
> `>>> A`: According to our experiment, using $\boldsymbol{I}^{'}$ in the text-attack stage indeed increases the ASR, which rises from 78.05 to 78.54. But there is a trade-off between the ASR rate and the perturbation degree.
>
> Using the clean image $\boldsymbol{I}$ as the input of the text-attack makes the perturbation degree of the images to be zero in this stage. When using the perturbed image \$\boldsymbol{I}^{'}\$ as the input, if the text attack fails, then \$\boldsymbol{I}^{'}\$ will be fed into the multimodal attack, which does not affect the perturbation degree of images. However, if the text attack can succeed even with the clean image $\boldsymbol{I}$, in such a case, using \$\boldsymbol{I}^{'}\$ will increase the perturbation degree.
> Specifically, we compute the average $L_2$-distance between all output and original images in the 8-bit color space. The average $L_2$-distance of using \$\boldsymbol{I}^{'}\$  and $\boldsymbol{I}$ in the text-attack stage are 11.69 and 11.23, respectively. So the unnecessary modification means the increase of the perturbation degree of images.
>
> `>>> W4` ***The query cost***\
> `>>> A`: Thanks for the advice.
>
> __*(1) Query Times*__
> Among all baselines evaluated in Table 1 of our paper, only text attack methods BERT-Attack (B\&A) and R\&R do queries. So we compare VLAttack with BERT-Attack as it achieves better ASR.
>
> The average number of queries on BERT-Attack and VLAttack are 8.61 and 4.99 *times/sample*, respectively.
> Even when the ICSA stage involves extra queries, VLAttack has a lower query number than BERT-Attack. This is because some
>  samples that can be directly obtained in the image-attack stage using BSA, and they only query once, thus avoiding queries in the text-attack and ICSA stages.
>
> __*(2) Computation Costs*__
> We also analyze the computation cost on the largest pre-trained model OFA. Specifically, we report the average speed in seconds for computing an adversarial sample on the VQAv2 dataset. The results are shown in the table below:
>
> |Method| SSP | BERT-Attack | Co-Attack | VLAttack |
>  |-|-|-|-|-|
> |Speed (s)|7.02 | 6.07 |18.46 |9.73|
>
> We can find that attacking a single modality (i.e., SSP and BERT-Attack) is faster than attacking two modalities (i.e., Co-Attack and VLAttack). Also, the speed of VLAttack is noticeably faster than that of Co-Attack.
> This is because Co-Attack requires adding both image and text perturbations for every sample, but VLAttack can achieve a successful attack by adding perturbations to a single modality only.
> In sum, with the outstanding attack performance, the acceptable generation speed further demonstrates the effectiveness of our proposed VLAttack.
>
> `>>> Q1` ***Explanation for Fig.7 second column***\
> `>>> A`: In referring expression comprehension task, a bounding box prediction is considered correct only if the Intersection Over Union (IOU) with the label is greater than 0.5.  Based on this metric, the bounding box prediction in Fig.7 of our paper is incorrect after the perturbation. Actually, this predicted bounding box doesn't encompass the person's head and unnecessarily includes the space above the head.
>
> Besides, substituting ''jeans'' with ''denim'' is reasonable, as it doesn't alter the original meaning of the sentence. In sum, this example stands for the claim that the VLAttack can alter the correctness of predictions by introducing minor perturbations to images and texts.
>
> `>>> Q2` ***Fine-tune the model on the perturbation dataset, and then conduct the transfer attack***\
> `>>> A`: Thanks. We conduct the experiment on the ViLT model through VQA dataset. We first fine-tune the pre-trained model on 5K (size of the perturbation dataset) samples and then use the fine-tuned model to produce adversarial samples. We adopt the same fine-tuning process as in ViLT[1], and the fine-tuning loss is a categorical cross-entropy function. When attacking, we enlarge the loss and optimize for 40 iterations like BSA.
> Results are shown in the table below. The new method is named Surrogate Attack (SA).
>
> |Method| SA | BSA | VLAttack |
>  |-| - | - |-|
> |ASR|24.39 | 65.20 |__78.05__ |
>
> In the table, SA performs poorly compared to BSA and VLAttack. This is because large VL models struggle to train well on limited data samples. Even when provided with clean samples, these surrogate models hardly make accurate predictions, leading to inferior ASR results.

---

> > ### Comment · Reviewer_zqVz · 2023-08-11
> >
> > Thank the authors for providing a detailed rebuttal with experiments. Since the source of transferability remains the main challenge of this line of research, I will keep the rating as borderline accept.

---

> > > ### Author Response · Authors · 2023-08-18
> > >
> > > Dear Reviewer zqVz,
> > >
> > > Thanks for your valuable comments and feedback on our rebuttal stage! We really appreciate your time and effort.
> > >
> > > Sincerely,
> > >
> > > Authors

---

### Official Review · Reviewer_AgmN · 2023-06-27

**Soundness:** 2 fair
**Presentation:** 3 good
**Contribution:** 3 good
**Rating:** 5
**Confidence:** 4

**Summary:**

In this paper, the authors explore the adversarial vulnerability in visual language models. Specifically, a block-wise similarity attack is proposed to generate adversarial image examples, and the BERT-attack method is used for generating the adversarial text examples. The image and text pairs are perturbed by an iterative method with cross-search.

**Strengths:**

1. This paper explores the black-box attack for vision language models, which is novel.

2. The proposed method can generate perturbed inputs to decrease the performance of the fine-tuned models on multi-modal and uni-modal datasets effectively.

**Weaknesses:**

1. One important criterion for a successful adversarial attack is that the perturbed image cannot be distinguished by humans, and the semantic meaning of the text adversarial examples should be the same as the original one. However, there needs to be more analysis of the semantic similarity between the generated adversarial examples and the original text. The perturbation of the images should be measured as well. Most importantly, the perturbation of the generated image-text pairs should be analyzed.

2. By applying the iterative way for generating adversarial image and text pairs, the attackers need to query the large vision language model more times. The time costs and the average query numbers for a successful attack should be provided.

**Questions:**

It is better to provide the analysis of the semantic similarity and the perturbation degree, as mentioned in Weaknesses 1.

Compared to other methods, could you provide a thorough analysis of the costs and average query times of the proposed framework?

**Limitations:**

I did not find a potential negative societal impact.

---

> ### Author Rebuttal · Authors · 2023-08-09
>
> Thank you for the valuable review. Our responses are addressed below.\
> `>>> Q1` ***It is better to provide the analysis of the semantic similarity and the perturbation degree, as mentioned in Weaknesses 1***\
> `>>> A`: Thanks for your constructive suggestions. We compare the semantic similarity score for the text modality. Following previous work [20][25], the score is obtained by first using the universal sentence encoder[44] to embed the output and original sentences into vectors. Then we compute the cosine similarity between the two vectors. We compute the average $L_2$-distance as the perturbation degree for the image modality between all output and original images in the 8-bit color space. The experiment is developed on the  ViLT model through the VQAv2 dataset, and the results are illustrated in the following table.
>
> |Modality| Metric | SSP (image-only)| Co-Attack | VLAttack|
> |----| ---- | ---- |---- |---- |
> |Text|Semantic Similarity | 1.00 |0.97 |0.99|
> |Image|$L_2$ Distance | 11.51 |11.38 |11.23|
>
> We can observe that SSP has the highest semantic similarity score since it only perturbs the image modality. Due to the limited length of text, Co-Attack and the proposed VLAttack also achieve higher semantic similarity. However, the proposed VLAttack performs better than the baseline Co-Attack.
> There are two reasons: (1) In our model design, we first perturb the image modality and then the text modality. If the image attack succeeds, we no longer need to perturb the text. (2) Even if the single-modality attack fails, during the multimodal attack, we will first use the text perturbation candidates with the highest semantic similarity to generate adversarial samples, which also helps VLAttack to maintain semantic similarity.
>
> Our method also shows advantages in terms of $L_{2}$ distance on image perturbations, where VLAttack outperforms both baselines.
> This advantage arises because VLAttack adopts clean images as input during the text attack stage. Consequently, the adversarial samples produced at this stage do not perturb the image, leading to a reduced average $L_2$ distance.
>
> `>>> Q2` ***Compared to other methods, could you provide a thorough analysis of the costs and average query times of the proposed framework?***\
> `>>> A`: Thanks for your comments.  We detail the analysis of query times and computation costs as follows.
>
> __*(1) Query Times*__
> The proposed VLAattack conducts queries in three phases: (1) During the image attack phase, a query is made after optimizing with BSA for $N_{s}=20$ times (Algorithm 1, line 5). (2) During the text attack phase, BERT-Attack is used for querying (Algorithm 1, line 11). (3) In the ICSA stage, queries are made for every $N_k$ step (Algorithm 1, line 19).
> Among all baselines evaluated in Table 1 of our paper, only text attack methods BERT-Attack (B\&A) and R\&R need queries. Thus, we compare VLAttack with BERT-Attack as it achieves better performance.
>
> The average number of queries on BERT-Attack and VLAttack are __8.61__ and __4.99__ *times/sample*, respectively.
> We can find that even when the ICSA stage involves additional queries, VLAttack has a lower average number of queries than BERT-Attack. This is attributed to a portion of the samples that can be directly obtained in the image-attack stage using BSA, and they only query once, thus avoiding queries in the text-attack stage and the ICSA stage.
>
> __*(2) Computation Costs*__
> We also analyze the average computation costs on the largest pre-trained model OFA. Specifically, we report the average speed in seconds for computing an adversarial sample on the VQAv2 dataset. The results are shown in the following table:
>
> |Method| SSP| BERT-Attack  | Co-Attack | VLAttack |
> |----| ---- | ---- |---- |---- |
> |Speed (s)|7.02 | 6.07 |18.46 |9.73|
>
> We can observe that attacking a single modality (i.e., SSP for image and Bert-Attack for text) is faster than perturbing two modalities (i.e., Co-Attack and VLAttack), which is reasonable. Besides, the speed of VLAttack is noticeably faster than that of Co-Attack.
> This is because Co-Attack requires adding both image and text perturbations for every sample, but our method can achieve a successful attack by adding perturbations to a single modality only.
> In sum, with the outstanding attack performance, the acceptable generation speed further demonstrates the effectiveness of our proposed VLAttack.

---

> > ### Comment · Reviewer_AgmN · 2023-08-20
> >
> > Thanks for your reply! The clarification regarding semantic similarity and costs addressed my concerns.

---

> > > ### Author Response · Authors · 2023-08-21
> > >
> > > Dear Reviewer AgmN,
> > >
> > > We are delighted that our response addressed your concerns. Thank you for your insightful comments and valuable feedback!
> > >
> > > Sincerely,
> > >
> > > Authors

---

### Official Review · Reviewer_NrLG · 2023-07-07

**Soundness:** 3 good
**Presentation:** 3 good
**Contribution:** 3 good
**Rating:** 6
**Confidence:** 4

**Summary:**

The paper presents VLAttack, which is a method for perturbing multimodal examples such that a multimodal model would get them wrong. VLAttack does not assume access to fine-tuned models but does assume access to foundation models that are used to create these downstream models. The authors argue that this level of access is reasonable because there are several real-world examples where the foundation model is available but downstream models are not. VLAttack makes models fail significantly more than similar adversarial attack algorithms across several datasets and tasks.

**Strengths:**

As far as I am aware, the algorithm is novel (especially the visual perturbation part). And the authors showcase that it outperforms similar algorithms in the sense that it does not assume access to the fine-tuned model (which could be unreasonable) and it makes models get more errors.

**Weaknesses:**

Major issues:

It is great that VLAttack makes the models get higher error rates than other approaches, but crucially it is only a good result if VLAttack does not actually change the ground truth labels. It seems that you do not validate this with, e.g. crowd workers. Unless I missed it, the only validation that you do is presenting case studies of a few individual examples. Did you do a systematic validation?

Less important (but still issues):

More motivation would be nice. What scenarios are you envisioning for why VLAttack raises a safety concern? Self-driving car failures, etc.?

Can you make these adversarial perturbations useful? For example, can you train a model on them to get better performance?

**Questions:**

I tried to frame everything in the weaknesses section as questions.

**Limitations:**

Limitations seem adequately addressed.

---

> ### Author Rebuttal · Authors · 2023-08-09
>
> Thank you for the valuable review. Our responses are addressed below.\
> `>>>Q1` ***Systematic validation***\
> `>>>A`:  Adding a human evaluation experiment significantly increases the reliability of the proposed attack method. We conducted the human evaluation experiments using the results output from the ViLT model on the VQAv2 dataset using Amazon Mechanical Turk (MTurk). The baseline that we choose is SSP since it outperforms others in our experiments, as shown in Table 1 in our paper.
>
> Specifically, we randomly sampled 650 examples and stored the generated answers after the attack. To validate the performance of both approaches, we send the original image-question pairs and the corresponding generated answer to the MTurk system. We provide three choice candidates to workers -- '' Definitely Wrong '', '' Not Sure '', and '' Definitely Correct ''. A successful attack means that the worker will label '' Definitely Wrong '' to the pair. To make the annotations more accurate and reliable, each pair is annotated by three workers, and we report the majority choice as the final human evaluation result. The statistics of human evaluation results are shown in the following table. We can observe that the proposed VLAttack still significantly outperforms the strongest baseline SSP.
>
> |Method| SSP|VLAttack|
>  |----| ---- | ---- |
> |Definitely Correct $\downarrow$|413 | __307__ |
> |Not Sure $\uparrow$|__58__ | 56 |
> |DefinitelyWrong $\uparrow$|179 | __287__ |
>
> `>>>Q2` ***More motivation would be nice.Can you make these adversarial perturbations useful? For example, can you train a model on them to get better performance?***\
> `>>>A`:  As mentioned in lines 21-27 of our paper, the `''pre-train $\&$ fine-tune'' paradigm has become a prevalent trend when training vision language models. The pre-trained VL models can be efficiently deployed in different scenarios and tasks after fine-tuning, such as virtual assistants [a] and robotic control [b]. Against this background, a key question raised in our paper is whether these publicly available pre-trained models are robust, which is also the primary motivation of our work.
>
> As a pioneering research, we believe it can further benefit various research fields, such as adversarial fine-tuning for more robust vision-language models. In fact, there has already been adversarial fine-tuning work on language models [c], but research on vision-language models remains unexplored. We believe that the proposed VLAttack can serve as significant guidance for future studies in this field.
>
> [a] Tu, T., et.al. Learning better visual dialog agents with pretrained visual-linguistic representation.CVPR 2021\
> [b] Yecheng Jason Ma., et.al.. LIV: Language-Image Representations and Rewards for Robotic Control. ICML 2023 \
> [c] Dong, X., et al. How should pre-trained language models be fine-tuned towards adversarial robustness?. NIPS 2021

---

> > ### Comment · Reviewer_NrLG · 2023-08-16
> >
> > Thanks for running that experiment. It is informative, and convinced me to change my rating above the "accept" threshold.

---

> > > ### Author Response · Authors · 2023-08-18
> > >
> > > Dear Reviewer NrLG,
> > >
> > > We are delighted that our response has addressed your concerns. Thank you for your constructive suggestions and positive feedback!
> > >
> > > Sincerely,
> > >
> > > Authors

---

### Official Review · Reviewer_3TAC · 2023-07-07

**Soundness:** 3 good
**Presentation:** 3 good
**Contribution:** 3 good
**Rating:** 7
**Confidence:** 4

**Summary:**

This paper focuses on adversarially attacking the multimodal finetuned models without getting access to the finetuned weights. By utilizing the activations and the parameters of the open-accessed pretrained model, this work proposes a method called VLATTACK for creating the adversarial attack samples for downstream tasks. The creating method first attempts to learn image perturbations via block-wise similarity attack (BSA) strategy. If not succeed, the text attack using BertAttack will be facilitated. If the two single-modal attack both fail, VLATTACK will use cross-search attack (ICSA) method to construct multimodal disrupted adversarial samples. Experimental results show the effectiveness of VLATTACK on various cross-modal (image-text) tasks and model structures. Some detailed analysis on the adversarial samples are also conducted.



**Strengths:**

1. The experimental results are obtained based on various downstream tasks and pretrained models (in different architectures), which demonstrate the generalization ability of VLATTACK.
2. The choice of pretrained models are open-sourced, making the results more reproducible.
3. The performance of adversarial attack is good, compared with other typical and previous attacking methods.
4. The case study and ablation study are both well-conducted, providing insights on how the VLATTACK can achieve better attacking performance.

**Weaknesses:**

Generally speaking, I think this paper is a good work on the topic of adversarial attacking. If the following issues are addressed, I think it will be much better:
1. Experiments or analysis on adversarial multimodal datasets: Some adversarially constructed datasets are proposed in previous works, based on the datasets used in this work, like Adversarial VQA. Does the created adversarial samples share some similarity with these datasets? I think some discussion or analysis on this research question can be performed and will be very welcomed.
2. Evaluation soundness: For VQA and captioning, an extra human evaluation on whether the predictions under attack are actually wrong will be much recommended, considering the automatic metrics may make false-positives.
3. Experiments on out-of-domain datasets: For OFA, we know the pretraining corpus include the downstream datasets (VQA, RefCOCO and MSCOCO), which may result in the adversarial samples easier to be obtained using the pretrained model. It would be very good if downstream datasets which not appear in pretraining OFA can be considered in experiments.

**Questions:**

How would the performance of VLATTACK be affected under different model size (like OFA-base vs OFA-large) and image resolution ($224$ vs $480$)? I will be very happy if some insights can be provided.

**Limitations:**

The authors have proposed some limitations in the appendix, including the text attacking methods, downstream task scope and explored model architectures, which I think is reasonable and leave space for future research.

---

> ### Author Rebuttal · Authors · 2023-08-09
>
> Thanks for the valuable review. Our responses are addressed below.\
> `>>> W1` ***Experiments or analysis on adversarial multimodal dataset***\
> `>>> A`: Some methods [a,b] focus on constructing adversarial datasets, but they significantly diverge from the nature of our research work. Specifically, these adversarial datasets usually consist of new samples generated through human annotation [d] (e.g., writing questions until models give wrong predictions) or machine synthesis [a] (e.g., removing objects from images). They do not have any perturbation budget constraints and are mainly used to test the robustness of multi-modal models.
>
> However, in our work, we aim to add noise to the clean data under strict perturbation constraints. The generated samples maintain similar appearances to the clean ones but can cause incorrect predictions in the downstream task model. This not only tests the robustness of the model but, more importantly, exposes potential security concerns on these pre-trained models.
>
> [a] Agarwal, V., et.al. (2020). Towards causal vqa: Revealing and reducing spurious correlations by invariant and covariant semantic editing. CVPR 2020.
>
> [b] Li, L., et.al. (2021). Adversarial vqa: A new benchmark for evaluating the robustness of vqa models. ICCV 2021.
>
> `>>> W2` ***Evaluation soundness***\
> `>>> A`: Thanks for your valuable comments and constructive suggestion. Adding a human evaluation experiment significantly increases the reliability of the proposed attack method. We conducted the human evaluation experiments using the results output from the ViLT model on the VQAv2 dataset using Amazon Mechanical Turk (MTurk). The baseline that we choose is SSP since it outperforms others in our experiments, as shown in Table 1 in our paper.
>
> Specifically, we randomly sampled 650 examples and stored the generated answers after the attack. To validate the performance of both approaches, we send the original image-question pairs and the corresponding generated answer to the MTurk system. We provide three choice candidates to workers -- `'' Definitely Correct '', '' Not Sure '', and '' Definitely Wrong ''. A successful attack means that the worker will label ``Definitely Wrong'' to the pair. To make the annotations more accurate and reliable, each pair is annotated by three workers, and we report the majority choice as the final human evaluation result. The statistics of human evaluation results are shown in the following table. We can observe that the proposed VLAttack still significantly outperforms the strongest baseline SSP.
>
> |Method| SSP| VLAttack|
> |----| ---- | ---- |
> |Definitely Correct $\downarrow$|413 | __307__ |
> Not Sure $\uparrow$|__58__| 56 |
> |Definitely Wrong $\uparrow$|179 | __287__ |
>
> `>>> W3` ***Experiments on out-of-domain datasets***\
> `>>> A`: Thanks for the advice. We do have such experiments, which are shown in Table 1 of our original paper. We conduct experiments on the SNLI-VE dataset for the visual entailment (VE) task, which is not involved in the pre-training process. From the results on the SNLI-VE dataset, we can observe that VLAttack achieves the best performance. Thus, even using the out-of-domain datasets, the proposed VLAttack is still effective.
>
> `>>> Q1` ***How would the performance of VLATTACK be affected under different model size
> (like OFA-base vs OFA-large) and image resolution (224 vs 480)?***\
> `>>> A`: Thanks for the constructive suggestions. These suggested experiments are interesting and will significantly improve the quality of our paper. Due to the limited time of the rebuttal stage, we only test the influence of model size and will put the results regarding the image resolution experiment in the final version.
>
> We evaluate different methods on the OFA-large model through the VQA task. The results are shown in the following table. We can observe that the proposed VLAttack achieves the best performance, but the results slightly decrease compared to those on OFA-base in Table 1 of our paper, which supports the claim that the model can be more robust with a larger parameter size.
>
> |Method| SSP|Co-Attack|VLAttack|
> |----| ---- | ---- | ---- |
> |ASR|33.82 | 39.41 |__75.44__|

---

> > ### Comment · Reviewer_3TAC · 2023-08-20
> >
> > Thank the authors for providing the detailed response. I will keep the rating as accept.

---

> > > ### Author Response · Authors · 2023-08-20
> > >
> > > Dear Reviewer 3TAC,
> > >
> > > Thanks for your positive feedback! We sincerely appreciate your effort and the acknowledgment of our paper!
> > >
> > > Sincerely,
> > >
> > > Authors

---

### Official Review · Reviewer_pd9a · 2023-07-07

**Soundness:** 3 good
**Presentation:** 4 excellent
**Contribution:** 2 fair
**Rating:** 5
**Confidence:** 4

**Summary:**

The authors propose a substitute black-box attack strategy called VLAttack to generate adversarial examples by perturbing both images and texts on pre-trained models and then transferring them to finetuned models.  At the image-modal level, they introduce a block-wise similarity attack (BSA) strategy to disrupt universal representations in the pre-trained model. At the multimodal level, they design an iterative cross-search attack (ICSA) to update adversarial image-text paris.

**Strengths:**

1. This paper focuses on an important question concerning the prevalent pretraining and finetuning paradigm with respect to adversarial robustness.

2. The core idea of the proposed attack is clearly presented.

3. The proposed VLAttack could potentially serve as a valuable baseline for examining the robustness of multimodal models.

**Weaknesses:**

Issues that affect my rating.

1. Lack of technique contributions. The question asked by the authors is quite valuable. However, the proposed attack strategy, VLAttack, seems trivial.  Regarding the text modality, VLAttack directly employs the existing BertAttack[20]. For the image attack,  the classical PGD attack with iteration $N_{s}=20$ is adapted. The idea of the proposed BSA loss Eq.2 is straightforward and similar to that of BadEncoder [a] (see BadEncoder's Eq.2~5), except BadEncoder focuses on the backdoor attack. The cross-modality attack should be the most important part, where the two modality attacks boost each other is excepted. However, the proposed "ICSA" seems directly concatenate the image and text attacks. Did the author have attempted to jointly optimize these two attacks?

[a] Jia, J., Liu, Y., & Gong, N. Z. (2022, May). Badencoder: Backdoor attacks to pre-trained encoders in self-supervised learning. In 2022 IEEE Symposium on Security and Privacy (SP) (pp. 2043-2059). IEEE.

2. Insufficient evaluation on popular multimodal models (CLIP, BLIP): I recommend that the authors extend their evaluation to include more prevalent multimodal models, such as CLIP or BLIP, as done in Co-attack[15] and BadEncoder[a]. This would enable readers and subsequent works to make more meaningful comparisons.

3. Ambiguity in the fine-tuned model settings: Considering the significant performance difference between VLAttack and baselines in Table 1, it is unclear how the target models are fine-tuned. Are the pre-trained model parameters fixed while training a task-specific head, or are all parameters fine-tuned throughout the pre-trained model? Providing clarity on this aspect is crucial for evaluating the impact of the proposed attack. Additional details should be included to enhance understanding.

4. Absence of essential ablation studies on BSA and ICSA: The effectiveness of BSA remains unclear, and a comparison with an attack on the pre-trained model's original loss could provide valuable insights. To showcase the efficacy of ICSA, a straightforward baseline could involve setting BSA's attack iteration $N_{s}=40$, without incorporating BERT-attack. The authors should consider conducting these ablation studies to better demonstrate their contributions.

**Questions:**

Please refer to the weakness section.

**Limitations:**

Yes

---

> ### Author Rebuttal · Authors · 2023-08-09
>
> Thank you for the valuable review. Our responses are addressed below.\
> `>>>Q1` ***Lack of technique contributions***\
> `>>>A`: The multimodal attack is a new research topic, and only one work Co-Attack has been proposed recently. However, its performance is even worse than state-of-the-art image attack approaches. To solve these issues, we designed a new multimodal attack approach named VLAttack, which can jointly optimize adversaries for both images and text. It first seeks potential adversarial candidates at the single-modality level. If unsuccessful, the outputs from single-modality attacks will be treated as initialization at the multi-modal attack level.
>
> In this work, we have two technical contributions. One is the new loss for the image attack, and the other is a novel cross-modality search strategy for the multimodal attack. In the single-modality attack, we propose a new block-wise attack loss to disrupt the features in the image encoder and transformer encoder of the pre-trained model. We directly apply BERT-Attack for the text modality due to the limited length of input text, as described in lines 189-193. Finally, we propose a novel cross-search strategy on top of the outputs from single-modality attacks, which iteratively perturbs image and text modalities further. Thus, we argue that the proposed VLAttack is simple, effective, and novel for the multimodal attack topic.
>
> `>>>Q2` ***Insufficient evaluation on popular multimodal models (CLIP, BLIP)***\
> `>>>A`: Thanks for the suggestion. We further extend our experiments on the CLIP model. The experiment is developed on the image classification task on the SVHN dataset [a]. Concretely, we use the image encoder of CLIP as the pre-trained model $F$ and then fine-tune $F$ on the SVHN dataset after adding a classification head. For the choices of image encoder of CLIP, we adopt ViT-B/16 and ResNet-50, denoted by CLIP-ViT-B/16 and CLIP-RN50. We test the attack performance using 1,000 correctly predicted samples.
> All results are illustrated in the following table. Since the task only accepts images as input, we compare our BSA with other baselines. As shown in the table, our proposed BSA still maintains the best ASR using different image encoder structures, clearly demonstrating its effectiveness.
> |Method| DR | SSP | FDA | BSA(VLAttack) |
> |----| ---- | ---- |---- |---- |
> |CLIP-ViT-B/16|7.90 | 4.60 |6.20 |__20.10__|
> |CLIP-RN50|70.70 | 76.20 |74.20 |__77.40__|
>
> [a] Goodfellow I J, et al. Multi-digit number recognition from street view imagery using deep convolutional neural networks[J]. arXiv preprint, 2013.
>
> `>>>Q3`  ***Ambiguity in the fine-tuned model settings***\
> `>>>A`: Thanks for your valuable comments. All task models are fully fine-tuned. In other words, all the model parameters are updated in the fine-tuning stage, but they are not accessible during the attacking stage. We treated them as black boxes. Thus, we do not have any knowledge about the fine-tuned model parameters during the attack process. We will add these details in the final version.
>
> `>>>Q4`:  ***Absence of essential ablation studies on BSA and ICSA***\
> `>>>A`: Thanks for your comments.
>
>  __*(1) Ablation study of BSA*__\
> To validate the effectiveness of BSA, we conduct an experiment to compare the original pre-trained loss with the proposed VLAttack on the ViLT model using the VQA task as an example. Since different pre-training tasks have their task-specific loss functions, in this experiment, we validate the masked language modeling (MLM) loss as the attacking objective for the VQA task.
>
> Given an image-text pair, the MLM loss randomly masks 15% of words in the text, and the training target is to recover these masked tokens. The attack objective is optimized through cross-entropy. In our implementation, we replace Eq. (2) in our paper with the MLM loss and reverse the target by adding perturbation on the image to make the pre-trained model incapable of recovering the masked tokes. When attacking, we optimize the above objective for 40 iterations like BSA. The results are illustrated in the following table.
>
> |Method|MLM | BSA | VLAttack |
> |----| ---- | ---- |---- |
> |ASR|25.40 | 65.20 |__78.05__|
>
> We can find that the performance using the original pre-trained MLM loss is far inferior to the proposed BSA and VLAttack methods. The reason is that in the attacking stage, the downstream tasks are fine-tuned with task-specific datasets, which also update the relations among tokens learned in the pertaining stage accordingly. However, if we still use the attack strategy learned in the pre-training stage, the change of token relations will make it fail to generate adversarial samples. Thus, the original MLM loss performs worse than our proposed VLAttack, even BSA.
>
> __*(2) Ablation study of ICSA*__\
> Thanks for pointing this out. We did have such baselines that involve setting BSA's attack iteration $N_s=40$, without incorporating BERT-attack, as shown in Figure 5 of Section 5.4. The experimental results show that our method equipped with ICSA (VLAttack) significantly outperforms the BSA method under the same iterations.

---

> > ### Comment · Reviewer_pd9a · 2023-08-16
> >
> > Thanks for the authors' response.
> >
> > **Question 1: Insufficient technical contributions** - The authors' responses do not fully address my concerns. It would be beneficial if the authors could directly answer my question: Have attempts been made to jointly optimize the text and image attacks? If so, why? If not, why not? Additionally, the authors might consider discussing and positioning their proposed Block-wise Similarity Attack (BSA) approach in comparison to the recently proposed BadEncoder [a] (see BadEncoder's Eq.2~5). Clarifying the distinction between these two methods could help to strengthen the contribution of this work.
> >
> > [a] Jia, J., Liu, Y., & Gong, N. Z. (2022, May). Badencoder: Backdoor attacks to pre-trained encoders in self-supervised learning. In 2022 IEEE Symposium on Security and Privacy (SP) (pp. 2043-2059). IEEE.
> >
> > **Question 2: Insufficient evaluation on popular multimodal models (CLIP, BLIP)**
> > I would still insist that it is important to augment the results in Table 1 with those from the CLIP and BLIP architectures, as done in Co-attack[15]. Omitting this step would make it challenging for readers to draw fair comparisons and accurately interpret the results.

---

> > > ### Author Response · Authors · 2023-08-18
> > >
> > > `>>> Q1-1`: Joint Optimization
> > >
> > > We have attempted to optimize both image and text perturbations jointly, but the performance of the joint optimization is worse than that of the proposed VLAttack.Toward the joint optimization, we modify the proposed BSA loss by adding a new loss term for the text encoder, and its format is similar to the image encoder and multimodal encoder, where we compare the block-wise similarity between the clean text input and the perturbed text input.
> > >
> > > At each attack step, we directly follow the current BSA design to generate image perturbations for the image modality. We then move to the text modality. Due to the discrete nature of text data, we follow existing work [21] using the word embedding space to generate the text perturbations. First, we use the modified loss to calculate the gradients on the text and then add them to the original word embeddings. The summation can be treated as the ideal word embeddings of the perturbed text. The next step is to map these continuous embeddings to discrete words. For each informative word in the original text, we use BERT to calculate its synonym set following [20]. We then use the word synonym substitution approach to replace the original words with their synonyms under the constraint listed in Eq. (1) in the original paper to guarantee the semantics of the new perturbation.
> > >
> > > The new image and text perturbations will be iteratively updated until $N$ steps with the above processes. The experimental results are shown in the following table:
> > > |Dataset| BSA| BSA-joint| VLAttack|
> > > |-|-|-|-|
> > > |VQAv2|65.20|66.48|__78.05__|
> > >
> > > BSA-joint denotes the joint optimization, which performs slightly better than the proposed BSA on the VQAv2 dataset but is still far inferior compared to VLAttack. This is because the input text $\mathbf{T}$ is relatively short, and frequently perturbing texts and strictly coupling the updates may lead to fluctuations in the gradient information. Therefore, joint optimization degrades the overall attack performance.
> > >
> > > `>>> Q1-2`: BSA v.s. BadEncoder
> > >
> > > We admit the overall idea of the proposed BSA and BadEncoder is similar, and they all use similarity as the optimization target. However, the proposed BSA is different from or better than BadEncoder.
> > >
> > > (1) BadEncoder (Eq.2--5) only utilizes the final output feature vectors from the whole encoder, ignoring the outputs from the intermediate layers/blocks. However, BSA calculates fine-grained similarity scores. As shown in Eq. (2) of our original paper, we distinguish the outputs from image and Transformer encoders. Such a design can modify the low-level vision features and the high-level cross-modal representations. In our setting, we attack fine-tuned models of a wide diversity of downstream tasks, including but not limited to image classification tasks like Badencoder. The parameters of these task-specific models are fully finetuned on distinct datasets, and the output representations of the encoder significantly change accordingly. Thus, instead of only attacking the output feature from the last layer like BadEncoder, perturbing each intermediate feature representation from each encoder and each block can enhance the attack performance. This statement is also verified in Section 5.4, where the ASR score of BSA is higher than only attacking a single encoder.
> > >
> > > (2) The motivation for adopting the cosine distance is different. In BadEncoder, CLIP uses cosine similarity as a loss function to calculate distances for positive/negative image text pairs, which is motivated by the pre-training strategy of CLIP. However, we adopt cosine similarity because the fine-grained token representations attend to each other in the inner product space, which is inspired by the mechanism design of the Transformer structure. This motivation is also illustrated in Lines 180-181 of our original paper.
> > >
> > > We will add these discussions in the final version.
> > >
> > > `>>> Q2`: CLIP and BLIP
> > >
> > > Thanks for the constructive suggestions. We agree that adding more experiments on CLIP and BLIP is beneficial for validating the contributions of our work more comprehensively. For the CLIP model, we will add the experiments through the image classification task, which has been discussed in the previous response on the SVHN dataset. We will add results using more datasets.
> > > For the BLIP model, we experiment with the VQA task of the BLIP model using the VQAv2 dataset. The proposed models still achieve better performance, as shown below:
> > > |Dataset|DR|SSP|FDA|BSA|B&A|R&R|Co-Attack|VLAttack|
> > > |-|-|-|-|-|-|-|-|-|
> > > |VQAv2|7.04|11.84|7.12|26.36|15.30|2.94|20.62|__45.64__|
> > >
> > > We will add all experimental results in our revised version to enhance the validation of model effectiveness.
> > >
> > > We sincerely appreciate your valuable feedback, which elevates the significance of our model's design, highlights its distinctions from existing approaches, and fortifies our work's overall quality. We hope our responses can adequately address your concerns.

---

> > > > ### Author Response · Authors · 2023-08-19
> > > >
> > > > Dear Reviewer __pd9a__,
> > > >
> > > > Thanks again for your insightful and constructive comments, which are significantly helpful in improving the quality of our paper. We kindly seed confirmation on whether our last response has adequately addressed your concerns. We would like to discuss further any valuable questions and concerns you have.
> > > >
> > > > Thanks,
> > > > Authors

---

> > > > ### Comment · Reviewer_pd9a · 2023-08-20
> > > >
> > > > After reviewing the authors' responses and additional experiments, my previous concerns have been largely addressed. The paper makes a good preliminary attempt at examining the robustness of multimodal models. Given this, I am inclined to upgrade my evaluation to 'borderline accept'.

---

> > > > > ### Author Response · Authors · 2023-08-20
> > > > >
> > > > > Dear Reviewer pd9a,
> > > > >
> > > > > We're glad that our response has addressed your concerns. Thank you for your effort and positive feedback!
> > > > >
> > > > > Sincerely,
> > > > >
> > > > > Authors

---

### Decision · Program_Chairs · 2023-09-21

**Decision:**

Accept (poster)

**Comment:**

The paper presents a preliminary exploration of multimodal model robustness. The experimental results span various downstream tasks and pretrained models, showcasing VLATTACK's generalization ability. The effectiveness of the adversarial attack in comparison to other methods is noteworthy. Both the case study and ablation study are well-executed, offering insights into how VLATTACK achieves superior attack performance.  Although the contribution might be seen as somewhat modest, the paper deserves a weak acceptance due to its demonstrated novel aspects and improved experimental validation.